# Non-Ergodic Alternating Proximal Augmented Lagrangian Algorithms with Optimal Rates

**Quoc Tran-Dinh**[*]

Department of Statistics and Operations Research, University of North Carolina at Chapel Hill
*Address:* Hanes Hall 333, UNC-Chapel Hill, NC27599, USA.
*Email:* quoctd@email.unc.edu

## Abstract

We develop two new non-ergodic alternating proximal augmented Lagrangian algorithms (NEAPAL) to solve a class of nonsmooth constrained convex optimization problems. Our approach relies on a novel combination of the augmented Lagrangian framework, alternating/linearization scheme, Nesterov's acceleration techniques, and adaptive strategy for parameters. Our algorithms have several new features compared to existing methods. Firstly, they have a Nesterov's acceleration step on the primal variables compared to the dual one in several methods in the literature. Secondly, they achieve non-ergodic optimal convergence rates under standard assumptions, i.e. an $\mathcal{O}\left(\frac{1}{k}\right)$ rate without any smoothness or strong convexity-type assumption, or an $\mathcal{O}\left(\frac{1}{k^2}\right)$ rate under only semi-strong convexity, where $k$ is the iteration counter. Thirdly, they preserve or have better per-iteration complexity compared to existing algorithms. Fourthly, they can be implemented in a parallel fashion. Finally, all the parameters are adaptively updated without heuristic tuning. We verify our algorithms on different numerical examples and compare them with some state-of-the-art methods.

## 1 Introduction

**Problem statement:** We consider the following nonsmooth constrained convex problem:

$$F^\star := \min_{z:=(x,y)\in\mathbb{R}^p} \left\{ F(z) := f(x) + \sum_{i=1}^m g_i(y_i) \ \text{s.t.} \ Ax + \sum_{i=1}^m B_i y_i = c \right\}, \tag{1}$$

where $f : \mathbb{R}^{\bar{p}} \to \mathbb{R} \cup \{+\infty\}$ and $g_i : \mathbb{R}^{p_i} \to \mathbb{R} \cup \{+\infty\}$ are proper, closed, and convex functions; $p := \bar{p} + \hat{p}$ with $\hat{p} := \sum_{i=1}^m p_i$; $A \in \mathbb{R}^{n\times\bar{p}}$, $B_i \in \mathbb{R}^{n\times p_i}$, and $c \in \mathbb{R}^n$ are given. Here, we also define $y := [y_1, \cdots, y_m]$ as a column vector, $g(y) := \sum_{i=1}^m g_i(y_i)$, and $By := \sum_{i=1}^m B_i y_i$. We often assume that we do not explicitly form matrices $A$ and $B_i$, but we can only compute $Ax$, $B_i y_i$ and their adjoints $A^\top\lambda$ and $B_i^\top\lambda$ for any given $x, y_i$, and $\lambda$ for $i = 1, \cdots, m$.

Problem (1) is sufficiently general to cope with many applications in different fields including machine learning, statistics, image/signal processing, and model predictive control. In particular, (1) covers convex empirical risk minimization, support vector machine, LASSO-type, matrix completion, compressive sensing problems as representative examples.

**Our approach:** Our approach relies on a novel combination of the augmented Lagrangian (AL) function and other classical and new techniques. First, we use AL as a merit function. Next, we incorporate an acceleration step (either Nesterov's momentum [17] or Tseng's accelerated variant [25]) into the primal steps. Then, we alternate the augmented Lagrangian primal subproblem into $x$ and $y$. We also linearize the $y_i$-subproblems and parallelize them to reduce per-iteration complexity. Finally, we incorporate with an adaptive strategy proposed in [23] to derive explicit update rules for algorithmic parameters. Our approach shares some similarities with the alternating direction method of multipliers (ADMM) and alternating minimization algorithm (AMA) but is essentially different from several aspects as will be discussed below.

**Our contribution:** Our contribution can be summarized as follows:

(a) We propose a novel algorithm called NEAPAL, Algorithm 1, to solve (1) under only convexity and strong duality assumptions. This algorithm can be viewed as a Nesterov's accelerated, alternating, linearizing, and parallel proximal AL method which alternates between $x$ and $y_i$, and linearizes and parallelizes the $y_i$-subproblems.

---

[*]This work is partly supported by the NSF-grant, DMS-1619884, USA.

(b) We prove an optimal $\mathcal{O}\left(\frac{1}{k}\right)$-rate of this algorithm in terms of $|F(z^k) - F^\star|$ and $\|Ax^k + By^k - c\|$. Our rate achieves at the last iterate (i.e. in a non-ergodic sense), while our per-iteration complexity is the same or even better than existing methods.

(c) When the problem (1) is semi-strongly convex, i.e. $f$ is non-strongly convex and $g$ is strongly convex, we develop a new NEAPAL variant, Algorithm 2, that achieves an optimal $\mathcal{O}\left(\frac{1}{k^2}\right)$-rate. This rate is either in a semi-ergodic (i.e. non-ergodic in $x$ and ergodic in $y$) sense or a non-ergodic sense. The non-ergodic rate just requires one more proximal operator of $g$. This variant also possesses the same parallel computation feature as in Algorithm 1.

From a practical point of view, Algorithm 1 has better per-iteration complexity than ADMM and AMA since the $y_i$-subproblems are linearized and parallelized. This per-iteration complexity is essentially the same as in primal-dual methods [3] when applying to solve composite convex problems with linear operators. When $f = 0$, we obtain fully parallel variants of Algorithms 1 and 2 which only require the proximal operators of $g_i$ and solve all the $y_i$-subproblems in parallel.

In terms of theory, Algorithm 1 achieves an optimal $\mathcal{O}\left(\frac{1}{k}\right)$-rate in a non-ergodic sense. Moreover, the dual step does not require averaging. Algorithm 2 only requires $F$ to be semi-strongly convex to achieve an optimal $\mathcal{O}\left(\frac{1}{k^2}\right)$-rate on the last iterate, which is weaker than that of the accelerated ADMM method in [11]. To our best knowledge, optimal rates at the last iterate have not been known yet in primal-dual methods such as in [3].[2] The $\mathcal{O}\left(\frac{1}{k^2}\right)$-rate is also achieved in [29] for accelerated ADMM, but Algorithm 2 remains essentially different from [29]. First, it combines different acceleration schemes for $x$ and $y$. Second, the convergence rate can achieve in either a non-ergodic or semi-ergodic sense. Third, the parameters are updated explicitly.

**Related work:** Our algorithms developed in this paper can be cast into the framework of augmented Lagrangian-type methods. In this context, we briefly review some notable and recent works which are most related to our methods. The augmented Lagrangian method was dated back from the work of Powell and Hessenberg in nonlinear programming in early 1970s. It soon became a powerful method to solve nonlinear optimization and constrained convex optimization problems. Alternatively, alternating methods were dated back from Von Neumann's work. Among these algorithms, AMA and ADMM are the most popular ones. ADMM can be viewed either as the dual variant of Douglas-Rachford's method [8, 15] or as an alternating variant of AL methods [1]. ADMM is widely used in practice, especially in signal and image processing. [2] provides a comprehensive survey of ADMM using in statistical learning. While the asymptotic convergence of ADMM has been long known, see, e.g., [8], its $\mathcal{O}\left(\frac{1}{k}\right)$-convergence rate seems to be first proved in [13, 16]. However, such a rate in [13] is achieved through a gap function in the framework of variational inequality and in an ergodic sense. The same $\mathcal{O}\left(\frac{1}{k}\right)$-non-ergodic rate was then proved in [12], but still on the sequence of differences $\{\|w^{k+1} - w^k\|^2\}$ combining both the primal and dual variables in $w$. Many other works also focus on theoretical aspects of ADMM by showing its $\mathcal{O}\left(\frac{1}{k}\right)$-convergence rate in the objective residual $|F(z^k) - F^\star|$ and feasibility gap $\|Ax^k + By^k - c\|$. Notable works include [6, 11, 20, 22]. Extensions to stochastic settings as well as multi-blocks formulations have also been intensively studied in the literature such as [4, 7]. Other researchers have attempted to optimize the rate of convergence in certain cases such as quadratic problems or using the theory of feedback control [10, 19].

In terms of algorithms, the main steps of ADMM remain the same in most of the existing research papers. Some modifications have been made for ADMM such as relaxation [6, 20, 22], or dual acceleration [11, 20]. Other extensions to Bregman distances and proximal settings remain essentially the same as the original version, see, e.g., [26]. Note that our algorithms can be cast into a primal-dual method such as [3, 23] rather than ADMM when solving composite problems with linear operators.

In terms of theory, most of existing results have shown an ergodic convergence rate of $\mathcal{O}\left(\frac{1}{k}\right)$ in either gap function or in both objective residual and constraint violation [5, 6, 11, 13, 20, 22, 27]. This rate has been shown to be optimal for ADMM-type methods under only convexity and strong duality in recent work [14, 28]. When one function $f$ or $g$ is strongly convex, one can achieve $\mathcal{O}\left(\frac{1}{k^2}\right)$ rate as shown in [29] but it is still on an averaging sequence. A recent work in [14] proposed a linearized ADMM variant using Nesterov's acceleration step and showed $\mathcal{O}\left(\frac{1}{k}\right)$-non-ergodic rate. This scheme is very similar to a special case of Algorithm 1. However, our scheme has a better per-iteration

complexity than [14] since it updates $y_i$ in parallel instead of alternating as in [14]. Besides, our analysis is much simpler than [14] which is extremely long and involves various parameters.

**Paper organization:** The rest of this paper is organized as follows. Section 2 recalls the dual problem of (1) and optimality condition. It also provides a key lemma for convergence analysis. Section 3 presents two new NEAPAL algorithms and analyzes their convergence rate. It also considers an extension. Section 4 provides some representative numerical examples.

**Notations:** We work on finite dimensional spaces $\mathbb{R}^p$ and $\mathbb{R}^n$, equipped with a standard inner product $\langle \cdot, \cdot \rangle$ and norm $\| \cdot \|$. Given a proper, closed, and convex function $f$, $\mathrm{dom}(f)$ denotes its domain, $\partial f(\cdot)$ is its subdifferential, $f^*(y) := \sup_x \{\langle y, x \rangle - f(x)\}$ is its Fenchel conjugate, and $\mathrm{prox}_{\gamma f}(x) := \mathrm{argmin}_u \{f(u) + \frac{1}{2\gamma}\|u - x\|^2\}$ is its proximal operator, where $\gamma > 0$. We say that $\mathrm{prox}_{\gamma f}$ is *tractably proximal* if it can be computed efficiently in a closed form or by a polynomial algorithm. Several tractable proximity functions can be found from the literature. We say that $f$ has $L_f$-Lipschitz gradient if it is differentiable, and its gradient $\nabla f$ is Lipschitz continuous with the Lipschitz constant $L_f \in [0, +\infty)$, $f$ is $\mu_f$-strongly convex if $f(\cdot) - \frac{\mu_f}{2}\| \cdot \|^2$ is convex, where $\mu_f > 0$ is its strong convexity parameter. For a given convex set $\mathcal{X}$, $\mathrm{ri}(\mathcal{X})$ denotes its relative interior. For a given matrix $A$, we denote $\|A\|$ its operator (or spectral) norm.

## 2 Duality theory, fundamental assumption, and optimality conditions

The Lagrange function associated with (1) is $\mathcal{L}(x, y, \lambda) := f(x) + g(y) - \langle Ax + By - c, \lambda \rangle$, where $\lambda$ is the vector of Lagrange multipliers. The dual function is defined as

$$d(\lambda) := \max_{(x,y) \in \mathrm{dom}(F)} \left\{ \langle Ax + By - c, \lambda \rangle - f(x) - g(y) \right\} = f^*(A^\top \lambda) + g^*(B^\top \lambda) - \langle c, \lambda \rangle,$$

where $\mathrm{dom}(F) := \mathrm{dom}(f) \times \mathrm{dom}(g)$, and $f^*$ and $g^*$ are the Fenchel conjugates of $f$ and $g$, respectively. The dual problem of (1) is

$$d^\star := \min_{\lambda \in \mathbb{R}^n} \left\{ d(\lambda) \equiv f^*(A^\top \lambda) + g^*(B^\top \lambda) - \langle c, \lambda \rangle \right\}. \tag{2}$$

We say that a point $(x^\star, y^\star, \lambda^\star) \in \mathrm{dom}(F) \times \mathbb{R}^n$ is a saddle point of the Lagrange function $\mathcal{L}$ if for all $(x, y) \in \mathrm{dom}(F)$, and $\lambda \in \mathbb{R}^n$, one has

$$\mathcal{L}(x^\star, y^\star, \lambda) \leq \mathcal{L}(x^\star, y^\star, \lambda^\star) \leq \mathcal{L}(x, y, \lambda^\star). \tag{3}$$

We denote by $\mathcal{S}^\star := \{(x^\star, y^\star, \lambda^\star)\}$ the set of saddle points of $\mathcal{L}$, by $\mathcal{Z}^\star := \{(x^\star, y^\star)\}$ the set of primal components of saddle points, and by $\Lambda^\star := \{\lambda^\star\}$ the set of corresponding multipliers.

In this paper, we rely on the following general assumption used in any primal-dual-type method.

**Assumption 2.1.** Both functions $f$ and $g$ are proper, closed, and convex. The set of saddle points $\mathcal{S}^\star$ of the Lagrange function $\mathcal{L}$ is nonempty, and the optimal value $F^\star$ is finite and is attainable at some $(x^\star, y^\star) \in \mathcal{Z}^\star$.

We assume that Assumption 2.1 holds throughout this paper without recalling it in the sequel.

The optimality condition (or the KKT condition) of (1) can be written as

$$0 \in \partial f(x^\star) - A^\top \lambda^\star, \quad 0 \in \partial g(y^\star) - B^\top \lambda^\star, \quad \text{and} \quad Ax^\star + By^\star = c. \tag{4}$$

Let us assume that the following Slater condition holds:

$$\mathrm{ri}(\mathrm{dom}(F)) \cap \{(x, y) \mid Ax + By = c\} \neq \emptyset.$$

Then, the optimality condition (4) is necessary and sufficient for the strong duality of (1) and (2) to hold, i.e., $F^\star + D^\star = 0$, and the dual solution is attainable and $\Lambda^\star$ is bounded, see, e.g., [1].

In practice, we can only find an approximation $\tilde{z}^\star := (\tilde{x}^\star, \tilde{y}^\star)$ to $z^\star$ of (1) in the following sense:

**Definition 2.1.** Given a tolerance $\varepsilon := (\varepsilon_p, \varepsilon_d) > 0$, we say that $\tilde{z}^\star := (\tilde{x}^\star, \tilde{y}^\star) \in \mathrm{dom}(F)$ is an $\varepsilon$-solution of (1) if $|F(\tilde{z}^\star) - F^\star| \leq \varepsilon_p$ and $\|A\tilde{x}^\star + B\tilde{y}^\star - c\| \leq \varepsilon_d$.

Let us define an augmented Lagrangian function $\mathcal{L}_\rho$ associated with the constrained problem (1) as

$$\mathcal{L}_\rho(z, \lambda) := f(x) + g(y) - \langle \lambda, Ax + By - c \rangle + \frac{\rho}{2} \|Ax + By - c\|^2, \tag{5}$$

where $z := (x, y)$, $\lambda$ is the corresponding multiplier, and $\rho > 0$ is a penalty parameter. The following lemma characterizes approximate solutions of (1) whose proof is in Supplementary Document A.

**Lemma 2.1.** *Let $S_\rho(z, \lambda) := \mathcal{L}_\rho(z, \lambda) - F^\star$ for $\mathcal{L}_\rho$ defined by (5). Then, for any $z = (x, y) \in \text{dom}(F)$ and $\lambda^\star \in \Lambda^\star$, we have*

$$|F(z) - F^\star| \leq \max\left\{ S_\rho(z, \lambda) + \frac{\|\lambda\|^2}{2\rho}, \frac{1}{\rho}\|\lambda^\star\| R_d \right\} \quad \text{and} \quad \|Ax + By - c\| \leq \frac{R_d}{\rho}, \quad (6)$$

*where $R_d := \|\lambda - \lambda^\star\| + \sqrt{\|\lambda - \lambda^\star\|^2 + 2\rho S_\rho(z, \lambda)}$ and $\|\lambda - \lambda^\star\|^2 + 2\rho S_\rho(z, \lambda) \geq 0$.*

Using Lemma 2.1, our goal is to generate a sequence $\{(z^k, \rho_k)\}$ such that $S_{\rho_k}(z^k, \hat{\lambda}^0)$ converges to zero. In this case, we obtain $z^k$ as an approximate solution of (1) in the sense of Definition 2.1.

## 3 Non-Ergodic Alternating Proximal Augmented Lagrangian Algorithms

We first propose a new primal-dual algorithm to solve nonsmooth and nonstrongly convex problems in (1). Then, we present another variant for the semi-strongly convex case. Finally, we extend our methods to the sum of smooth and nonsmooth objectives.

### 3.1 NEAPAL for nonstrongly convex case

The classical augmented Lagrangian method minimizes the augmented Lagrangian function $\mathcal{L}_\rho$ in (5) over $x$ and $y$ altogether, which is often difficult. Our methods alternate between $x$ and $y$ to break the non-separability of the augmented term $\frac{\rho}{2}\|Ax + By - c\|^2$. Therefore, at each iteration $k$, given $\hat{z}^k := (\hat{x}^k, \hat{y}^k) \in \text{dom}(F)$, $\hat{\lambda}^k \in \mathbb{R}^n$, $\rho_k > 0$, and $\gamma_k \geq 0$, we define the $x$-subproblem as

$$\mathcal{S}_{\gamma_k}(\hat{z}^k, \hat{\lambda}^k; \rho_k) := \arg\min_{x \in \text{dom}(f)} \left\{ f(x) - \langle \hat{\lambda}^k, Ax \rangle + \frac{\rho_k}{2}\|Ax + B\hat{y}^k - c\|^2 + \frac{\gamma_k}{2}\|x - \hat{x}^k\|^2 \right\}. \quad (7)$$

If $\gamma_k > 0$, then (7) is well-defined and has unique solution. If $\gamma_k = 0$, then we need to assume that (7) has optimal solution but not necessarily unique. For the $y$-subproblem, we linearize the augmented term to make use of proximal operators of $g$. We also incorporate Nesterov's accelerated steps [18] into these primal subproblems. In summary, our algorithm is presented in Algorithm 1, which we call a Non-Ergodic Alternating Proximal Augmented Lagrangian (NEAPAL) method.

---

**Algorithm 1** (*Non-Ergodic Alternating Proximal Augmented Lagrangian Algorithm* (NEAPAL))

---

1: **Initialization:** Choose $z^0 := (x^0, y^0) \in \text{dom}(F)$, $\hat{\lambda}^0 \in \mathbb{R}^n$, $\rho_0 > 0$, and $\gamma_0 \geq 0$. Set $\tilde{z}^0 := z^0$.

2: **For** $k := 0$ **to** $k_{\max}$ **perform**

3:     (*Parameter update*) $\tau_k := \frac{1}{k+1}$, $\rho_k := \rho_0(k+1)$, $\beta_k := 2\rho_0 L_B(k+1)$, and $\eta_k := \frac{\rho_0}{2}$.

4:     (*Acceleration step*) $\hat{z}^k := (1 - \tau_k)z^k + \tau_k \tilde{z}^k$ with $z = (x, y)$.

5:     (*x-update*) $x^{k+1} := \mathcal{S}_{\gamma_k}(\hat{z}^k, \hat{\lambda}^k; \rho_k)$ by solving (7) and $r^k := Ax^{k+1} + B\hat{y}^k - c$.

6:     (*Parallel y-update*) **For** $i = 1$ **to** $m$ **update** $y_i^{k+1} := \text{prox}_{\frac{g_i}{\beta_k}}\left(\hat{y}_i^k - \frac{1}{\beta_k}B_i^\top(\rho_k r^k - \hat{\lambda}^k)\right)$.

7:     (*Momentum step*) $\tilde{z}^{k+1} := \tilde{z}^k + \frac{1}{\tau_k}(z^{k+1} - \hat{z}^k)$.

8:     (*Dual step*) $\hat{\lambda}^{k+1} := \hat{\lambda}^k - \eta_k(A\tilde{x}^{k+1} + B\tilde{y}^{k+1} - c)$.

9:     (*$\gamma$-update*) Choose $0 \leq \gamma_{k+1} \leq \left(\frac{k+2}{k+1}\right)\gamma_k$ if necessary.

10: **End for**

---

The parameter $L_B$ in Algorithm 1 can be chosen as $L_B := \|B\|^2$, or $L_B := m \max\left\{\|B_i\|^2 \mid 1 \leq i \leq m\right\}$. Moreover, we have a flexibility to choose $\rho_0$ and $\gamma_0$. For example, we can fix $\gamma_0 > 0$ to make sure (7) is well-defined. But if $A = \mathbb{I}$, the identity operator, or $A$ is orthogonal, then we should choose $\gamma_0 = 0$.

Combining Step 4 and Step 7, we can show that the per-iteration complexity of Algorithm 1 is dominated by the subproblem (7) at Step 5, one proximal operator of $g$, one matrix vector-multiplication $(Ax, By)$, and one adjoint $B^\top \lambda$. Hence, the per-iteration complexity of Algorithm 1 is better than that of standard ADMM [2]. We also observe the following additional features of Algorithm 1.

- Firstly, the subproblem (7) not only admits a unique solution, but it is also strongly convex. Hence, if we use first-order methods to solve it, then we obtain a linear convergence rate. In particular, if $A = \mathbb{I}$ or $A$ is orthonormal, then we can choose $\gamma_0 = 0$, and (7) reduces to the proximal operator of $f$, i.e.

$$\mathcal{S}_0(\hat{z}^k, \hat{\lambda}^k; \rho_k) := \text{prox}_{f/\rho_k}\left(A^\top(c - B\hat{y}^k - \rho_k^{-1}\hat{\lambda}^k)\right).$$

- Secondly, we directly incorporate Nesterov's accelerated steps into the primal variables instead of the dual one as in [11, 20]. We can eliminate $\tilde{z}^k$, and update $\hat{z}^{k+1} := z^{k+1} + \frac{k}{k+2}(z^{k+1} - z^k)$. In this case, the dual variable $\hat{\lambda}^k$ can be updated as

$$\hat{\lambda}^{k+1} := \hat{\lambda}^k - \frac{\eta_k}{\tau_k}\left(Ax^{k+1} + By^{k+1} - c - (1 - \tau_k)(Ax^k + By^k - c)\right).$$

  This dual update collapses to the one in classical AL methods such as AMA and ADMM, and their variants when $\tau_k = 1$ is fixed in all iterations.
- Thirdly, the parameters $\rho_k$ and $\beta_k$ are increasingly updated with the same rate of $\mathcal{O}(k)$, and $\gamma_k$ can be increasing, decreasing, or fixed. Moreover, while the penalty parameter $\rho_k$ is updated at each iteration, the step-size $\eta_k$ in the dual step remains fixed.
- Fourthly, we can use different parameters $\beta_k^i$ for each $y_i$-subproblem for $i = 1, \cdots, m$. In this case, we can update $\beta_k^i$ based on $L_{B_i} := m\|B_i\|^2$ for each component $i$.
- Finally, if $f = 0$, then we can remove the $x$-subproblem in Algorithm 1 to obtain a parallel variant of this algorithm. In this case, if we use different $\beta_k^i$, then they can be updated as $\beta_k^i := 2L_{B_i}(k+1)$. The convergence analysis of this variant requires some slight changes.

The convergence of Algorithm 1 is stated in the following theorem whose proof can be found in Supplementary Document B.

**Theorem 3.1.** *Let $\{z^k\}$ be the sequence generated by Algorithm 1. Then, for any $k \geq 1$, we have*

$$|F(z^k) - F^\star| \leq \frac{1}{2\rho_0 k}\max\left\{\rho_0 R_0^2 + \|\hat{\lambda}^0\|^2, 2R_d\|\lambda^\star\|\right\} \quad and \quad \|Ax^k + By^k - c\| \leq \frac{R_d}{\rho_0 k}, \quad (8)$$

*where $R_0^2 := \gamma_0\|x^0 - x^\star\|^2 + 2\rho_0 L_B\|y^0 - y^\star\|^2$ and $R_d := \|\hat{\lambda}^0 - \lambda^\star\| + \sqrt{\|\hat{\lambda}^0 - \lambda^\star\|^2 + \rho_0 R_0^2}$.*

*Consequently, the sequence of the last iterates $\{z^k\}$ globally converges to a solution $z^\star$ of (1) at a non-ergodic optimal $\mathcal{O}\left(\frac{1}{k}\right)$-rate, i.e., $|F(z^k) - F^\star| \leq \mathcal{O}\left(\frac{1}{k}\right)$ and $\|Ax^k + By^k - c\| \leq \mathcal{O}\left(\frac{1}{k}\right)$.*

### 3.2 NEAPAL for semi-strongly convex case

Now, we propose a new variant of Algorithm 1 that can exploit the semi-strong convexity of $F$. Without loss of generality, we assume that $g_i$ is strongly convex with the convexity parameter $\mu_{g_i} > 0$ for all $i = 1, \cdots, m$. In this case $g(y) = \sum_{i=1}^m g_i(y_i)$ is also strongly convex with the parameter $\mu_g := \min\{\mu_{g_i} \mid 1 \leq i \leq m\} > 0$.

To exploit the strong convexity of $g$, we apply Tseng's accelerated scheme in [25] to the $y$-subproblem, while using Nesterov's momentum idea [17] for the $x$-subproblem to keep the non-ergodic convergence on $\{x^k\}$. The complete algorithm is described in Algorithm 2.

---

**Algorithm 2** (scvx-NEAPAL for solving (1) with strongly convex objective term $g$)

---

1: **Initialization:** Choose $z^0 := (x^0, y^0) \in \text{dom}(F)$, $\hat{\lambda}^0 \in \mathbb{R}^n$, $\rho_0 \in \left(0, \frac{\mu_g}{4L_B}\right]$, and $\gamma_0 \geq 0$.

2:     Set $\tau_0 := 1$ and $\tilde{z}^0 := z^0$.

3: **For** $k := 0$ to $k_{\max}$ **perform**

4:     (*Parameter update*) Set $\rho_k := \frac{\rho_0}{\tau_k^2}$, $\gamma_k := \gamma_0$, $\beta_k := 2L_B\rho_k$, and $\eta_k := \frac{\rho_k\tau_k}{2}$.

5:     (*Accelerated step*) $\hat{z}^k := (1 - \tau_k)z^k + \tau_k\tilde{z}^k$ with $z = (x, y)$.

6:     (*x-update*) $x^{k+1} := \mathcal{S}_{\gamma_k}(\hat{z}^k, \hat{\lambda}^k; \rho_k)$ by solving (7) and $r^k := Ax^{k+1} + B\hat{y}^k - c$.

7:     (*x-momentum step*) $\tilde{x}^{k+1} := \tilde{x}^k + \frac{1}{\tau_k}(x^{k+1} - \hat{x}^k)$.

8:     (*Parallel $\tilde{y}$-update*) For $i = 1$ to $m$, update $\tilde{y}_i^{k+1} := \text{prox}_{\frac{g_i}{\tau_k\beta_k}}\left(\tilde{y}_i^k - \frac{1}{\tau_k\beta_k}B_i^\top(\rho_k r^k - \hat{\lambda}^k)\right)$.

9:     (*Dual step*) $\hat{\lambda}^{k+1} := \hat{\lambda}^k - \eta_k(A\tilde{x}^{k+1} + B\tilde{y}^{k+1} - c)$.

10:    (*Parallel $y$-update*) For $i = 1$ to $m$, update $y_i^{k+1}$ using **one** of the following two **options**:

$$\begin{bmatrix} y_i^{k+1} := (1 - \tau_k)y_i^k + \tau_k\tilde{y}_i^{k+1} & \textbf{(Option 1: Averaging step)} \\ y_i^{k+1} := \text{prox}_{\frac{g_i}{\rho_k L_B}}\left(\hat{y}_i^k - \frac{1}{\rho_k L_B}B_i^\top(\rho_k r^k - \hat{\lambda}^0)\right) & \textbf{(Option 2: Proximal step).} \end{bmatrix}$$

11:    ($\tau$-update) $\tau_{k+1} := \frac{1}{2}\tau_k\left(\sqrt{\tau_k^2 + 4} - \tau_k\right)$.

12: **End for**

---

The parameter $L_B$ is chosen as in Algorithm 1, and $\mu_g := \min\{\mu_{g_i} \mid 1 \le i \le m\}$ in Algorithm 2. We can replace the choice of $\rho$ in Algorithm 2 by $0 < \rho_0 \le \min\left\{\frac{\mu_{g_i}}{4L_{B_i}} \mid 1 \le i \le m\right\}$, where $L_{B_i} := \|B_i\|^2$. Before analyzing the convergence of Algorithm 2, we make the following remarks:

   (a) Firstly, Algorithm 2 linearizes the $y$-subproblem to reduce the per-iteration complexity. This step relies on Tseng's accelerated variant in [25] instead of Nesterov's optimal scheme [17] as in Algorithm 1. Hence, it uses two different options at Step 10 to form $y^{k+1}$.

   (b) Secondly, if $y^{k+1}$ is updated using **Option 1**, then one can take a weighted averaging step on $y^k$ without incurring extra cost. The **Option 2** at Step 10 requires one additional $\text{prox}_g$ but can avoid averaging on $y^k$ as in **Option 1**.

   (c) Thirdly, we can eliminate all parameters $\gamma_k$, $\beta_k$, and $\eta_k$ in Algorithm 2 so that it has only two parameters $\tau_k$ and $\rho_0$ that need to be updated and initialized, respectively.

The following theorem proves convergence of Algorithm 2 (*cf.* Supplementary Document D).

**Theorem 3.2.** *Assume that $g_i$ is $\mu_{g_i}$-strongly convex with $\mu_{g_i} > 0$ for all $i = 1, \cdots, m$, but $f$ is not necessarily strongly convex. Let $\{z^k\}$ be generated by Algorithm 2. Then, the following bounds hold:*

$$|F(z^k) - F^\star| \le \frac{2}{\rho_0(k+1)^2}\left\{\rho_0 R_0^2 + \|\hat{\lambda}^0\|^2, 2R_d\|\lambda^\star\|\right\} \quad and \quad \|Ax^k + By^k - c\| \le \frac{4R_d}{\rho_0(k+1)^2}, \quad (9)$$

*where $R_0^2 := \gamma_0\|x^0 - x^\star\|^2 + 2\rho_0 L_B\|y^0 - y^\star\|^2$ and $R_d := \|\hat{\lambda}^0 - \lambda^\star\| + \sqrt{\|\hat{\lambda}^0 - \lambda^\star\|^2 + 2\rho_0 R_0^2}$.*

*Consequently, $\{z^k\}$ globally converges to $z^\star$ at $\mathcal{O}\left(\frac{1}{k^2}\right)$-rate either in a semi-ergodic sense (i.e. non-ergodic in $x^k$ and ergodic in $y^k$) if **Option 1** is chosen, or a non-ergodic sense if **Option 2** is chosen, i.e., $|F(z^k) - F^\star| \le \mathcal{O}\left(\frac{1}{k^2}\right)$ and $\|Ax^k + By^k - c\| \le \mathcal{O}\left(\frac{1}{k^2}\right)$.*

### 3.3 Extension to the sum of smooth and nonsmooth objective functions

We can consider (1) with $F(z) := f(x) + \hat{f}(x) + \sum_{i=1}^m \left[g_i(y_i) + \hat{g}_i(y_i)\right]$, where $\hat{f}$ and $\hat{g}_i$ are smooth with $L_{\hat{f}}$- and $L_{\hat{g}_i}$-Lipschitz gradients, respectively. In this case, the $x$- and $y_i$-subproblems in Algorithm 1 can be replaced respectively by

$$\begin{cases} x^{k+1} := \underset{x}{\arg\min}\left\{f(x) + \langle \nabla\hat{f}(\hat{x}^k) - A^\top\hat{\lambda}^k, x - \hat{x}^k\rangle + \frac{\rho_k}{2}\|Ax + B\hat{y}^k - c\|^2 + \frac{\hat{\gamma}_k}{2}\|x - \hat{x}^k\|^2\right\}, \\ y_i^{k+1} := \underset{y_i}{\arg\min}\left\{g_i(y_i) + \langle \nabla\hat{g}_i(\hat{y}_i^k) + B_i^\top\left(\rho_k r^k - \hat{\lambda}^k\right), y_i - \hat{y}_i^k\rangle + \frac{\hat{\beta}_k^i}{2}\|y_i - \hat{y}_i^k\|^2\right\}, \end{cases}$$

where $\hat{\gamma}_k := \gamma_k L_A + L_{\hat{f}}$ and $\hat{\beta}_k^i := \beta_k L_{B_i} + L_{\hat{g}_i}$ for $i = 1, \cdots, m$. We can also modify Algorithm 2 and its convergence guarantee to handle this case, but we omit the details here.

## 4 Numerical experiments

We provide some numerical examples to illustrate our algorithms. More examples can be found in Supplementary Document E. All the experiments are implemented in Matlab R2014b, running on a MacBook Pro. Retina, 2.7GHz Intel Core i5 with 16Gb RAM.

### 4.1 Square-root LASSO and Square-root Elastic-net

We consider the following square-root elastic-net problem as a modification of the model in [30]:

$$F^\star := \min_{y \in \mathbb{R}^{p_2}}\left\{F(y) := \|By - c\|_2 + \frac{\kappa_1}{2}\|y\|_2^2 + \kappa_2\|y\|_1\right\}, \quad (10)$$

where $B \in \mathbb{R}^{n \times \hat{p}}$, $c \in \mathbb{R}^n$, and $\kappa_1 \ge 0$ and $\kappa_2 > 0$ are two regularization parameters. If $\kappa_1 = 0$, then (10) reduces to the well-known square-root LASSO model which is fully nonsmooth.

**Square-root LASSO Problem:** We first compare our algorithms with state-of-the-art methods on the square-root LASSO problem. Since this problem is fully nonsmooth and non-strongly convex, we implement three candidates to compare: ASGARD [23] and its restarting variant, and Chambolle-Pock's method [3]. For ASGARD, we use the same setting as in [23], and for Chambolle-Pock's (CP) method, we use step-sizes $\sigma = \tau = \|B\|^{-1}$ and $\theta = 1$. In Algorithm 1, we choose $\rho_0 := \frac{\|\lambda^\star\|}{\|B\|\|y^0 - y^\star\|}$ as suggested by Theorem 3.1 to trade-off the objective residual and feasibility gap, where $(x^\star, \lambda^\star)$ is computed by MOSEK up to the best accuracy. In Algorithm 2, we set $\rho_0 := \frac{\mu_g}{4\|B\|^2}$ as suggested by our theory, where $\mu_g := 0.1 \times \sigma_{\min}(B)$ as a guess for the restricted strong convexity parameter.

We generate $B$ randomly using standard Gaussian distribution without or with $50\%$ correlated columns. Then, we normalize $B$ to get unit norm columns. We generate $c$ as $c := By^\natural + \mathcal{N}(0,\sigma)$, where $y^\natural$ is a $s$-sparse vector, and $\sigma = 0$ (i.e. without noise) and $\sigma = 10^{-3}$ (i.e. with noise). In square-root LASSO, we set $\kappa_1 = 0$ and $\kappa_2 = 0.055$ which gives us reasonable results close to $y^\natural$.

We run these algorithms on two problem instances, where $(n, p, s) = (700, 2000, 100)$, and the results are plotted in Figure 1. Here, `NEAPAL` is Algorithm 1, `scvx-NEAPAL` is Algorithm 2, `NEAPAL-par` is the parallel variant of Algorithm 1 by setting $f = 0$ and $g_1(y_1) = \|By - c\|_2$ and $g_2(y_2) := \kappa_2\|y\|_1$, and `ASGARD-rs` is the restarting-ASGARD [23], and `avg-CP` is the averaging sequence of `CP`.

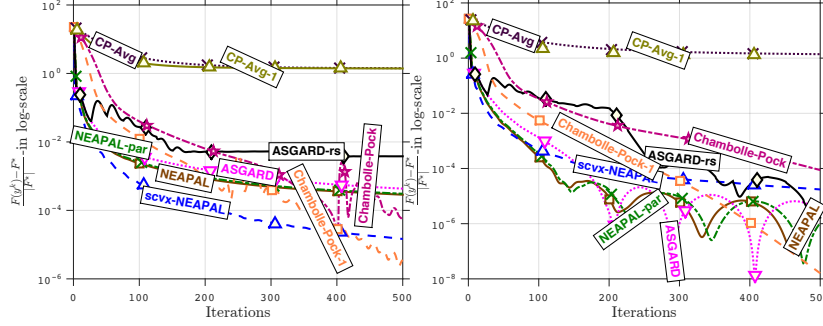

Figure 1: Convergence behavior on the relative objective residuals of 6 algorithms for the square-root LASSO problem (10) after 500 iterations. Left: Without noise; Right: With noise and $50\%$ correlated columns.

We can observe from Figure 1 that Algorithm 1 and its parallel variant has similar performance and are comparable with `ASGARD`. Algorithm 2 also performs well compared to other methods. It works better than Chambolle-Pock's method (`CP`) in early iterations, but becomes slower in late iterations. `ASGARD-rs` does not perform well due to the lack of strong convexity. While the last iterate of `CP` shows a great progress, its averaging sequence, where we have convergence rate guarantee is very slow in both cases: standard case and the case where the stepsize $\tau = 1$.

**Square-root Elastic-net Problems:** Now, we consider the case $\kappa_1 = 0.01 > 0$ in (10), which is called the square-root elastic-net problem. Our data is generated as in square-root LASSO. In this case, Algorithm 2 and Chambolle-Pock's method with strong convexity are used. The results of these algorithms and non-strongly convex variants are plotted in Figure 2.

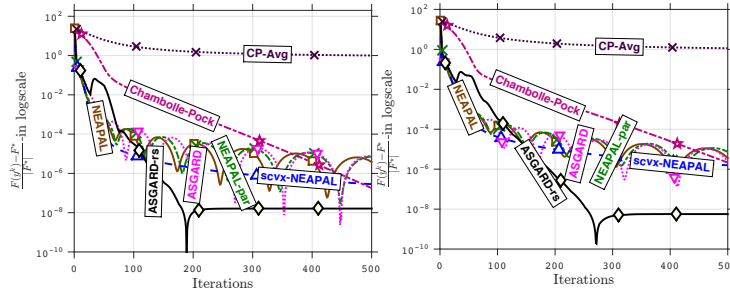

Figure 2: Convergence behavior on the relative objective residuals of 6 algorithms for the square-root elastic-net problem (10) after 500 iterations. Left: Without noise; Right: With noise and $50\%$ correlated columns.

`NEAPAL`, `NEAPAL-par`, and `scvx-NEAPAL` all work well in this example. They are all comparable with `ASGARD`. `CP` makes a slow progress in early iterations, but reaches better accuracy at the end. `ASGARD-rs` is the best due to the strong convexity of the problem. However, it does not have a theoretical guarantee. Again, the averaging sequence of `CP` is the slowest one.

## 4.2 Low-rank matrix recovery with square-root loss

We consider a low-rank matrix recovery problem with square-root loss, which can be considered as a penalized formulation of the model in [21]:

$$F^\star := \min_{Y \in \mathbb{R}^{m \times q}} \left\{ F(Y) := \|\mathcal{B}(Y) - c\|_2 + \lambda\|Y\|_* \right\}, \tag{11}$$

where $\|\cdot\|_*$ is a nuclear norm, $\mathcal{B} : \mathbb{R}^{m \times q} \to \mathbb{R}^n$ is a linear operator, $c \in \mathbb{R}^n$ is a given observed vector, and $\lambda > 0$ is a penalty parameter. By letting $z := (x, Y)$, $F(z) := \|x\|_2 + \lambda\|Y\|_*$ and $-x + \mathcal{B}(Y) = c$, we can reformulate (11) into (1).

To avoid complex subproblems of ADMM, we reformulate (11) into the following one:

$$\min_{x,Y,Z} \left\{ \|x\|_2 + \lambda \|Z\|_* \mid -x + \mathcal{B}(Y) = c, \ Y - Z = 0 \right\},$$

by introducing two auxiliary variables $x := \mathcal{B}(Y) - c$ and $Z := Y$. The main computation at each iteration of ADMM includes $\mathrm{prox}_{\|\cdot\|_*}$, $\mathcal{B}(Y)$, $\mathcal{B}^*(x)$, and the solution of $(\mathbb{I} + \mathcal{B}^*\mathcal{B})(Y) = e^k$, where $e^k$ is a residual term computed at each iteration. Since $\mathcal{B}$ and $\mathcal{B}^*$ are given in operators, we apply a preconditioned conjugate gradient (PCG) method to solve it. We warm-start PCG and terminate it with a tolerance of $10^{-5}$ or a maximum of 30 iterations. We tune the penalty parameter $\rho$ in ADMM for our test and find that $\rho = 0.25$ works best. We call this variant "Tuned-ADMM".

We test 3 algorithms: Algorithm 1, ASGARD [23], and Tuned-ADMM on 5 logo images: IBM, EPFL, MIT, TUM, and UNC. These images are pre-processed to get low-rank forms of 45, 59, 6, 7 and 56, respectively. The measurement $c$ is generated as $c := \mathcal{B}(Y^\natural) + \mathcal{N}(0, 10^{-3} \max |Y_{ij}^\natural|)$ with Gaussian noise, where $Y^\natural$ is the original image, and $\mathcal{B}$ is a linear operator formed by a subsampled-FFT with 25% of sampling rate. We run 3 algorithms with 200 iterations, and the results are given in Table 1.

Table 1: The results and performance of 3 algorithms on 5 Logo images of size $256 \times 256$.

| Name | ASGARD [23] | | | | | Algorithm 1 (NEAPAL) | | | | | Tuned-ADMM | | | | |
|---|---|---|---|---|---|---|---|---|---|---|---|---|---|---|---|
| | Time | Error | $F(Y^k)$ | PSNR rank | Res | Time | Error | $F(Y^k)$ | PSNR rank | Res | Time | Error | $F(Y^k)$ | PSNR rank | Res |
| IBM | 8.0 | 0.0615 | 0.293 | 72.4 34 | 0.107 | 8.5 | 0.0604 | 0.297 | 72.4 34 | 0.107 | 12.7 | 0.0615 | 0.293 | 72.4 34 | 0.107 |
| EPFL | 8.2 | 0.0830 | 0.414 | 69.8 56 | 0.108 | 8.1 | 0.0803 | 0.426 | 69.8 56 | 0.108 | 17.2 | 0.0830 | 0.414 | 69.8 56 | 0.108 |
| MIT | 7.9 | 0.0501 | 0.348 | 74.2 6 | 0.102 | 7.5 | 0.0485 | 0.349 | 74.2 6 | 0.102 | 15.9 | 0.0502 | 0.348 | 74.2 6 | 0.102 |
| TUM | 7.5 | 0.0382 | 0.266 | 76.5 49 | 0.087 | 7.6 | 0.0390 | 0.277 | 76.5 49 | 0.087 | 20.1 | 0.0384 | 0.267 | 76.5 49 | 0.087 |
| UNC | 8.3 | 0.0611 | 0.283 | 72.5 42 | 0.112 | 7.7 | 0.0596 | 0.287 | 72.5 42 | 0.112 | 14.7 | 0.0611 | 0.283 | 72.5 42 | 0.112 |

The results in Table 1 show that ASGARD and NEAPAL work well and are comparable with Tuned-ADMM. However, NEAPAL and ASGARD are faster than ADMM due to the PCG loop for solving the linear system. The recovered results of two images: TUM and MIT are shown in Figure 3. Except for TUM, three algorithms produce low-rank solutions as expected, and their PSNR

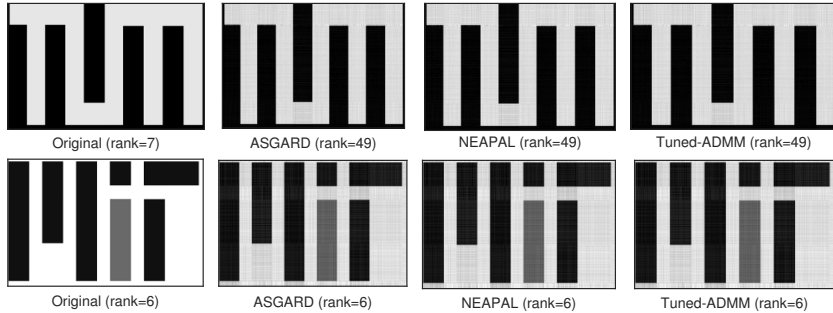

Original (rank=7)   ASGARD (rank=49)   NEAPAL (rank=49)   Tuned-ADMM (rank=49)

Original (rank=6)   ASGARD (rank=6)   NEAPAL (rank=6)   Tuned-ADMM (rank=6)

Figure 3: The low-rank recovery from three algorithms on two loge images: TUM and MIT.

(peak-signal-to-noise-ratio) is consistent. Moreover, $\texttt{Error} := \frac{\|Y^k - Y^\natural\|_F}{\|Y^\natural\|_F}$ showing the relative error between $Y^k$ and the original image $Y^\natural$ is small in all cases.

## 5  Conclusion

We have proposed two novel primal-dual algorithms to solve a broad class of nonsmooth constrained convex optimization problems that have the following features. They offer the same or better per-iteration complexity as existing methods such as AMA or ADMM. They achieve optimal convergence rates in non-ergodic sense (i.e., in the last iterates) on the objective residual and feasibility violation, which are important in sparse and low-rank optimization as well as in image processing. They can be implemented in both sequential and parallel manner. The dual update step in Algorithms 1 and 2 can be viewed as the dual step in relaxed augmented Lagrangian-based methods, where the step-size is different from the penalty parameter. Our future research is to develop new variants of Algorithms 1 and 2 such as coordinate descent, stochastic primal-dual, and asynchronous parallel algorithms. We also plan to investigate connection of our methods to primal-dual first-order methods such as primal-dual hybrid gradient and projective and other splitting methods.

## Footnotes

[2] In [14], a non-ergodic rate is obtained, but the algorithm is essentially different. However, a non-ergodic optimal rate of first-order methods for solving (1) was perhaps first proved in [24].

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
