[Supplementary Material]


# Non-Ergodic Alternating Proximal Augmented Lagrangian Algorithms with Optimal Rates

## A Properties of Augmented Lagrangian Function and Optimality Bounds

In this section, we investigate some properties of the augmented Lagrangian function $\mathcal{L}_\rho$ in (5).

### 1.1 Properties of the augmented Lagrangian function

Let us recall the augmented Lagrangian function $\mathcal{L}_\rho$ in (5) associated with problem (1). To investigate its properties, we define the following two functions:

$$\psi_\rho(u, \lambda) := \tfrac{\rho}{2}\|u\|^2 - \langle \lambda, u \rangle, \quad \text{and} \quad \phi_\rho(z, \lambda) := \psi_\rho(Ax + By - c, \lambda). \tag{12}$$

Since $\nabla_u \psi_\rho(u, \hat{\lambda}) = \rho u - \hat{\lambda}$ is $\rho$-Lipschitz continuous in $u$ for any given $\hat{\lambda} \in \mathbb{R}^n$, it is obvious that

$$
\begin{aligned}
\psi_\rho(u_+, \hat{\lambda}) &\le \psi_\rho(u, \hat{\lambda}) + \langle \nabla_u \psi_\rho(u, \hat{\lambda}), u_+ - u \rangle + \tfrac{\rho}{2}\|u_+ - u\|^2 \\
\psi_\rho(u_+, \hat{\lambda}) &\ge \psi_\rho(u, \hat{\lambda}) + \langle \nabla_u \psi_\rho(u, \hat{\lambda}), u_+ - u \rangle + \tfrac{1}{2\rho}\|\nabla_u \psi_\rho(u_+, \hat{\lambda}) - \nabla_u \psi_\rho(u, \hat{\lambda})\|^2,
\end{aligned}
\tag{13}
$$

for any $u_+, u \in \mathbb{R}^n$, see, e.g., [18].

Given $\hat{z}^{k+1} := (x^{k+1}, \hat{y}^k) \in \text{dom}(F)$ and $\hat{\lambda}^k \in \mathbb{R}^n$, we also define the following linear function:

$$\ell_\rho^k(z) := \phi_\rho(\hat{z}^{k+1}, \hat{\lambda}^k) + \langle \nabla_x \phi_\rho(\hat{z}^{k+1}, \hat{\lambda}^k), x - x^{k+1} \rangle + \langle \nabla_y \phi_\rho(\hat{z}_{k+1}, \hat{\lambda}^k), y - \hat{y}^k \rangle. \tag{14}$$

If we define $s^k := Ax^k + By^k - c$ and $\hat{s}^{k+1} := Ax^{k+1} + B\hat{y}^k - c$, then using the definition of $\ell_\rho^k$ and $\phi_\rho$, we can easily show that

$$
\begin{aligned}
\ell_\rho^k(z) &= \phi_\rho(z, \hat{\lambda}^k) - \tfrac{\rho}{2}\|A(x - x^{k+1}) + B(y - \hat{y}^k)\|^2, \quad \forall z \in \text{dom}(F), \\
\ell_\rho^k(z^\star) &= -\tfrac{\rho}{2}\|\hat{s}^{k+1}\|^2 \quad \text{and} \quad \ell_\rho^k(z^k) = \phi_\rho(z^k, \hat{\lambda}^k) - \tfrac{\rho}{2}\|s^k - \hat{s}^{k+1}\|^2,
\end{aligned}
\tag{15}
$$

where $z^\star \in \mathcal{Z}^\star$ is any solution of (1).

For any matrix $B := [B_1, \cdots, B_m]$ concatenated from $m$ matrices $B_i$ for $i = 1, \cdots, m$, we define $L_B := \|B\|^2$ and $\bar{L}_B := m \cdot \max\{\|B_i\|^2 \mid 1 \le i \le m\}$, where $\|B\|$ and $\|B_i\|$ is the operator norms of $B$ and $B_i$, respectively. For any $d = [d_1, \cdots, d_m] \in \mathbb{R}^{\hat{p}}$, we can easily show that

$$\|Bd\|^2 = \|\sum_{i=1}^m B_i d_i\|^2 \le \|B\|^2 \|d\|^2 \le m \sum_{i=1}^m \|B_i\|^2 \|d_i\|^2 \le \bar{L}_B \|d\|^2. \tag{16}$$

By the definition of $\phi_\rho$, using (14), (15), and (16), for any $(x, y) \in \text{dom}(F)$, $\hat{y} \in \text{dom}(g)$, and $\hat{\lambda} \in \mathbb{R}^n$, we can derive

$$\phi_\rho(x, y, \hat{\lambda}) - \phi_\rho(x, \hat{y}, \hat{\lambda}) - \langle \nabla_y \phi_\rho(x, \hat{y}, \hat{\lambda}), y - \hat{y} \rangle = \frac{\rho}{2}\|B(y - \hat{y})\|^2.$$

Hence, by (16), we can show that

$$\phi_\rho(x, y, \hat{\lambda}) - \phi_\rho(x, \hat{y}, \hat{\lambda}) - \langle \nabla_y \phi_\rho(x, \hat{y}, \hat{\lambda}), y - \hat{y} \rangle \le \frac{\rho L_B}{2}\|y - \hat{y}\|^2 \le \frac{\rho \bar{L}_B}{2}\|y - \hat{y}\|^2. \tag{17}$$

### 1.2 The proof of Lemma 2.1: Approximate optimal solutions of (1)

For any $z \in \text{dom}(F)$, we have $F^\star = \mathcal{L}(z^\star, \lambda^\star) \le \mathcal{L}(z, \lambda^\star) = F(z) - \langle \lambda^\star, Ax + By - c \rangle$. Using the definition of $S_\rho(\cdot)$, we obtain

$$S_\rho(z, \lambda) + \langle \lambda, Ax + By - c \rangle - \frac{\rho}{2}\|Ax + By - c\|^2 = F(z) - F(z^\star) \ge \langle \lambda^\star, Ax + By - c \rangle. \tag{18}$$

This inequality implies

$$\tfrac{\rho}{2}\|Ax + By - c\|^2 - \|\lambda - \lambda^\star\|\|Ax + By - c\| - S_\rho(z, \lambda) \le 0, \tag{19}$$

which leads to

$$2\rho S_\rho(z,\lambda) + \|\lambda - \lambda^\star\|^2 \geq \rho^2\|Ax + By - c\|^2 - 2\rho\|\lambda - \lambda^\star\|\|Ax + By - c\| + \|\lambda - \lambda^\star\|^2$$
$$= [\rho\|Ax + By - c\| - \|\lambda - \lambda^\star\|]^2 \geq 0.$$

From from (19), we also have $\|Ax + By - c\| \leq \frac{1}{\rho}\left[\|\lambda - \lambda^\star\| + \sqrt{\|\lambda - \lambda^\star\|^2 + 2\rho S_\rho(z,\lambda)}\right]$ by solving a quadratic inequation. This is the second inequality of (6).

Next, from (18), we have

$$F(z) - F^\star \leq S_\rho(z,\lambda) - \frac{\rho}{2}\|Ax + By - c\|^2 + \|\lambda\|\|Ax + By - c\|$$
$$\leq S_\rho(z,\lambda) - \frac{\rho}{2}\left[\|Ax + By - c\| - \frac{\|\lambda\|}{\rho}\right]^2 + \frac{\|\lambda\|^2}{2\rho}$$
$$\leq S_\rho(z,\lambda) + \frac{\|\lambda\|^2}{2\rho}.$$

Using the Cauchy-Schwarz inequality, it follows from $F^\star \leq F(z) - \langle\lambda^\star, Ax + By - c\rangle$ that $-\|\lambda^\star\|\|Ax + By - c\| \leq F(z) - F^\star$. Combining these two inequalities and the second estimate of (6), we obtain the first estimate of (6). $\qquad\square$

# B  Convergence analysis of Algorithm 1

Lemma B.1 and Lemma B.2 below are key to analyze the convergence of Algorithm 1.

**Lemma B.1.** *Assume that $\mathcal{L}_\rho$ is defined by (5), and $\ell^k_{\rho_k}$ is defined by (14). Let $z^{k+1}$ be computed by Algorithm 1. Then, for any $z \in \mathrm{dom}(F)$, we have*

$$\mathcal{L}_{\rho_k}(z^{k+1}, \hat{\lambda}^k) \leq F(z) + \ell^k_{\rho_k}(z) + \gamma_k\langle x^{k+1} - \hat{x}^k, x - \hat{x}^k\rangle - \gamma_k\|x^{k+1} - \hat{x}^k\|^2 \tag{20}$$
$$+ \beta_k\langle y^{k+1} - \hat{y}^k, y - \hat{y}^k\rangle - \frac{(2\beta_k - \rho_k L_B)}{2}\|y^{k+1} - \hat{y}^k\|^2.$$

*Proof.* Using (17) with $\rho = \rho_k$, $(x,y) = (x^{k+1}, y^{k+1}) = z^{k+1}$, $(x,\hat{y}) = (x^{k+1}, \hat{y}^k) = \hat{z}^{k+1}$, and $\hat{\lambda} = \hat{\lambda}^k$, we have

$$\phi_{\rho_k}(z^{k+1}, \hat{\lambda}^k) \leq \phi_{\rho_k}(\hat{z}^{k+1}, \hat{\lambda}^k) + \langle\nabla_y\phi_{\rho_k}(\hat{z}^{k+1}, \hat{\lambda}^k), y^{k+1} - \hat{y}^k\rangle + \frac{\rho_k L_B}{2}\|y^{k+1} - \hat{y}^k\|^2. \tag{21}$$

Next, using again $\phi_\rho$ from (12), we can write down the optimality condition of the $x$-subproblem at Step 5 and the $y_i$-subproblem at Step 6 of Algorithm 1 as follows:

$$\begin{cases} 0 = \nabla f(x^{k+1}) + \nabla_x\phi_{\rho_k}(\hat{z}^{k+1}, \hat{\lambda}^k) + \gamma_k(x^{k+1} - \hat{x}^k), & \nabla f(x^{k+1}) \in \partial f(x^{k+1}), \\ 0 = \nabla g_i(y_i^{k+1}) + \nabla_{y_i}\phi_{\rho_k}(\hat{z}^{k+1}, \hat{\lambda}^k) + \beta_k(y_i^{k+1} - \hat{y}_i^k), & \nabla g_i(y_i^{k+1}) \in \partial g_i(y_i^{k+1}). \end{cases} \tag{22}$$

Using the convexity of $f$ and $g$, for any $x \in \mathrm{dom}(f)$ and $y \in \mathrm{dom}(g)$, we have

$$f(x^{k+1}) \leq f(x) + \langle\nabla f(x^{k+1}), x^{k+1} - x\rangle, \quad \nabla f(x^{k+1}) \in \partial f(x^{k+1}), \tag{23}$$
$$g(y^{k+1}) \leq g(y) + \langle\nabla g(y^{k+1}), y^{k+1} - y\rangle, \quad \nabla g(y^{k+1}) \in \partial g(y^{k+1}).$$

Combining (21), (22), and (23), and then using the definition (5) of $\mathcal{L}_\rho$, for any $z = (x,y) \in \mathrm{dom}(F)$, we can derive that

$$\mathcal{L}_{\rho_k}(z^{k+1}, \hat{\lambda}^k) = f(x^{k+1}) + g(y^{k+1}) + \phi_{\rho_k}(z^{k+1}, \hat{\lambda}^k)$$

$$\overset{(21),(23)}{\leq} f(x) + \langle\nabla f(x^{k+1}), x^{k+1} - x\rangle + g(y) + \langle\nabla g(y^{k+1}), y^{k+1} - y\rangle$$
$$+ \phi_{\rho_k}(\hat{z}^{k+1}, \hat{\lambda}^k) + \langle\nabla_y\phi_{\rho_k}(\hat{z}^{k+1}, \hat{\lambda}^k), y^{k+1} - \hat{y}^k\rangle + \frac{\rho_k L_B}{2}\|y^{k+1} - \hat{y}^k\|^2$$

$$\overset{(22)}{\leq} F(z) + \phi_{\rho_k}(\hat{z}^{k+1}, \hat{\lambda}^k) + \langle\nabla_x\phi_{\rho_k}(\hat{z}^{k+1}, \hat{\lambda}^k), x - x^{k+1}\rangle + \langle\nabla_y\phi_{\rho_k}(\hat{z}^{k+1}, \hat{\lambda}^k), y - \hat{y}^k\rangle$$
$$+ \gamma_k\langle\hat{x}^k - x^{k+1}, x^{k+1} - x\rangle + \beta_k\langle\hat{y}^k - y^{k+1}, y^{k+1} - y\rangle + \frac{\rho_k L_B}{2}\|y^{k+1} - \hat{y}^k\|^2$$

$$\overset{(14)}{=} F(z) + \ell^k_{\rho_k}(z) + \gamma_k\langle x^{k+1} - \hat{x}^k, x - \hat{x}^k\rangle - \gamma_k\|x^{k+1} - \hat{x}^k\|^2$$
$$+ \beta_k\langle y^{k+1} - \hat{y}^k, y - \hat{y}^k\rangle - \frac{(2\beta_k - \rho_k L_B)}{2}\|y^{k+1} - \hat{y}^k\|^2,$$

which is exactly (20). $\qquad\square$

**Lemma B.2.** *Let $(z^k, \hat{\lambda}^k, z^{k+1}, \tilde{z}^{k+1})$ be generated by Algorithm 1. Then, for any $\lambda \in \mathbb{R}^n$, if $0 \leq 2\eta_k \leq \rho_k \tau_k$, then one has*

$$
\mathcal{L}_{\rho_k}(z^{k+1}, \lambda) \leq (1-\tau_k)\mathcal{L}_{\rho_{k-1}}(z^k, \lambda) + \tau_k F(z^\star) + \tfrac{\gamma_k \tau_k^2}{2}\left[\|\tilde{x}^k - x^\star\|^2 - \|\tilde{x}^{k+1} - x^\star\|^2\right]
$$
$$
+ \tfrac{\beta_k \tau_k^2}{2}\left[\|\tilde{y}^k - y^\star\|^2 - \|\tilde{y}^{k+1} - y^\star\|^2\right] + \tfrac{\tau_k}{2\eta_k}\left[\|\hat{\lambda}^k - \lambda\|^2 - \|\hat{\lambda}^{k+1} - \lambda\|^2\right] \qquad (24)
$$
$$
- \tfrac{(\beta_k - 2\rho_k L_B)}{2}\|y^{k+1} - \hat{y}^k\|^2 - \tfrac{(1-\tau_k)}{2}\left[\rho_{k-1} - \rho_k(1-\tau_k)\right]\|s^k\|^2,
$$

*where $\tau_k \in [0,1]$, and $\rho_k$, $\beta_k$, $\gamma_k$, and $\eta_k$ are positive parameters, and $s^k := Ax^k + By^k - c$.*

*Proof.* Using (20) with $z = z^k$ and $z = z^\star$, respectively, and then using (15), we obtain

$$
\mathcal{L}_{\rho_k}(z^{k+1}, \hat{\lambda}^k) \overset{(15)}{\leq} \mathcal{L}_{\rho_k}(z^k, \hat{\lambda}^k) - \tfrac{\rho_k}{2}\|s^k - \hat{s}^{k+1}\|^2 + \gamma_k\langle x^{k+1} - \hat{x}^k, x^k - \hat{x}^k\rangle
$$
$$
- \gamma_k \|x^{k+1} - \hat{x}^k\|^2 + \beta_k\langle y^{k+1} - \hat{y}^k, y^k - \hat{y}^k\rangle - \tfrac{(2\beta_k - \rho_k L_B)}{2}\|y^{k+1} - \hat{y}^k\|^2,
$$

$$
\mathcal{L}_{\rho_k}(z^{k+1}, \hat{\lambda}^k) \overset{(15)}{\leq} F(z^\star) - \tfrac{\rho_k}{2}\|\hat{s}^{k+1}\|^2 + \gamma_k\langle x^{k+1} - \hat{x}^k, x^\star - \hat{x}^k\rangle - \gamma_k\|x^{k+1} - \hat{x}^k\|^2
$$
$$
+ \beta_k\langle y^{k+1} - \hat{y}^k, y^\star - \hat{y}^k\rangle - \tfrac{(2\beta_k - \rho_k L_B)}{2}\|y^{k+1} - \hat{y}^k\|^2.
$$

Here, $s^k := Ax^k + By^k - c$ and $\hat{s}^{k+1} := Ax^{k+1} + B\hat{y}^k - c$. Multiplying the first inequality by $(1-\tau_k) \in [0,1]$ and the second one by $\tau_k \in [0,1]$ and summing up the results, and then using the fact that $\mathcal{L}_{\rho_k}(z^k, \hat{\lambda}^k) = \mathcal{L}_{\rho_{k-1}}(z^k, \hat{\lambda}^k) + \tfrac{(\rho_k - \rho_{k-1})}{2}\|s^k\|^2$, we can estimate

$$
\mathcal{L}_{\rho_k}(z^{k+1}, \hat{\lambda}^k) \leq (1-\tau_k)\mathcal{L}_{\rho_k}(z^k, \hat{\lambda}^k) + \tau_k F(z^\star) - \tfrac{(1-\tau_k)\rho_k}{2}\|s^k - \hat{s}^{k+1}\|^2 - \tfrac{\tau_k \rho_k}{2}\|\hat{s}^{k+1}\|^2
$$
$$
+ \gamma_k \tau_k\langle x^{k+1} - \hat{x}^k, x^\star - \tilde{x}^k\rangle - \gamma_k\|x^{k+1} - \hat{x}^k\|^2 + \beta_k \tau_k\langle y^{k+1} - \hat{y}^k, y^\star - \tilde{y}^k\rangle
$$
$$
- \tfrac{\beta_k}{2}\|y^{k+1} - \hat{y}^k\|^2 - \tfrac{(\beta_k - \rho_k L_B)}{2}\|y^{k+1} - \hat{y}^k\|^2
$$
$$
= (1-\tau_k)\mathcal{L}_{\rho_{k-1}}(z^k, \hat{\lambda}^k) + \tau_k F(z^\star) - \tfrac{\gamma_k}{2}\|x^{k+1} - \hat{x}^k\|^2 - \tfrac{(\beta_k - \rho_k L_B)\tau_k^2}{2}\|\tilde{y}^{k+1} - \tilde{y}^k\|^2
$$
$$
+ \tfrac{\gamma_k \tau_k^2}{2}\left[\|\tilde{x}^k - x^\star\|^2 - \|\tilde{x}^{k+1} - x^\star\|^2\right] + \tfrac{\beta_k \tau_k^2}{2}\left[\|\tilde{y}^k - y^\star\|^2 - \|\tilde{y}^{k+1} - y^\star\|^2\right]
$$
$$
- \tfrac{(1-\tau_k)\rho_k}{2}\|s^k - \hat{s}^{k+1}\|^2 - \tfrac{\tau_k \rho_k}{2}\|\hat{s}^{k+1}\|^2 + \tfrac{(1-\tau_k)(\rho_k - \rho_{k-1})}{2}\|s^k\|^2. \qquad (25)
$$

Here, we use $\tau_k \tilde{x}^k = \hat{x}^k - (1-\tau_k)x^k$, $\tau_k \tilde{y}^k = \hat{y}^k - (1-\tau_k)y^k$, $\tau_k(\tilde{x}^{k+1} - \tilde{x}^k) = x^{k+1} - \hat{x}^k$, $\tau_k(\tilde{y}^{k+1} - \tilde{y}^k) = y^{k+1} - \hat{y}^k$, and an elementary expression $2\langle a, b\rangle - \|a\|^2 = \|a - b\|^2 - \|b\|^2$.

Now, let $\tilde{s}^{k+1/2} := A\tilde{x}^{k+1} + B\tilde{y}^k - c$. Then, it is trivial to estimate the quantity $\mathcal{T}_k$ below

$$
\mathcal{T}_k := \tfrac{(1-\tau_k)\rho_k}{2}\|s^k - \hat{s}^{k+1}\|^2 + \tfrac{\tau_k \rho_k}{2}\|\hat{s}^{k+1}\|^2 - \tfrac{(1-\tau_k)(\rho_k - \rho_{k-1})}{2}\|s^k\|^2
$$
$$
= \tfrac{\rho_k}{2}\|\hat{s}^{k+1} - (1-\tau_k)s^k\|^2 + \tfrac{(1-\tau_k)}{2}\left[\rho_{k-1} - \rho_k(1-\tau_k)\right]\|s^k\|^2 \qquad (26)
$$
$$
= \tfrac{\rho_k \tau_k^2}{2}\|\tilde{s}^{k+1/2}\|^2 + \tfrac{(1-\tau_k)}{2}\left[\rho_{k-1} - \rho_k(1-\tau_k)\right]\|s^k\|^2.
$$

Here, we use the fact that $\hat{s}^{k+1} - (1-\tau_k)s^k = Ax^{k+1} + B\hat{y}^k - c - (1-\tau_k)(Ax^k + By^k - c) = \tau_k(A\tilde{x}^{k+1} + B\tilde{y}^k - c) = \tau_k \tilde{s}^{k+1/2}$.

Using the relation $\mathcal{L}_\rho(z, \lambda) = \mathcal{L}_\rho(z, \hat{\lambda}) + \langle\hat{\lambda} - \lambda, Ax + By - c\rangle$ from (5), $z^{k+1} - (1-\tau_k)z^k = \tau_k \tilde{z}^{k+1}$, and (26), we can further derive from (25) for any $\lambda \in \mathbb{R}^n$ that

$$
\mathcal{L}_{\rho_k}(z^{k+1}, \lambda) \leq (1-\tau_k)\mathcal{L}_{\rho_{k-1}}(z^k, \lambda) + \tau_k F(z^\star) - \tfrac{(1-\tau_k)}{2}\left[\rho_{k-1} - \rho_k(1-\tau_k)\right]\|s^k\|^2
$$
$$
+ \tfrac{\gamma_k \tau_k^2}{2}\left[\|\tilde{x}^k - x^\star\|^2 - \|\tilde{x}^{k+1} - x^\star\|^2\right] + \tfrac{\beta_k \tau_k^2}{2}\left[\|\tilde{y}^k - y^\star\|^2 - \|\tilde{y}^{k+1} - y^\star\|^2\right] \qquad (27)
$$
$$
- \tfrac{\gamma_k}{2}\|x^{k+1} - \hat{x}^k\|^2 - \tfrac{(\beta_k - \rho_k L_B)\tau_k^2}{2}\|\tilde{y}^{k+1} - \tilde{y}^k\|^2
$$
$$
+ \tau_k\langle\hat{\lambda}^k - \lambda, A\tilde{x}^{k+1} + B\tilde{y}^{k+1} - c\rangle - \tfrac{\rho_k \tau_k^2}{2}\|\tilde{s}^{k+1/2}\|^2.
$$

Let $\tilde{s}^{k+1} := A\tilde{x}^{k+1} + B\tilde{y}^{k+1} - c$. From the update rule $\hat{\lambda}^{k+1} := \hat{\lambda}^k - \eta_k(A\tilde{x}^{k+1} + B\tilde{y}^{k+1} - c) = \hat{\lambda}^k - \eta_k \tilde{s}^{k+1}$, if we define $M_k := \tau_k\langle\hat{\lambda}^k - \lambda, A\tilde{x}^{k+1} + B\tilde{y}^{k+1} - c\rangle$, then we can estimate $M_k$ as

$$
M_k = \tfrac{\tau_k}{\eta_k}\langle\hat{\lambda}^k - \lambda, \hat{\lambda}^k - \hat{\lambda}^{k+1}\rangle = \tfrac{\tau_k}{2\eta_k}\left[\|\hat{\lambda}^k - \lambda\|^2 - \|\hat{\lambda}^{k+1} - \lambda\|^2\right] + \tfrac{\tau_k}{2\eta_k}\|\hat{\lambda}^k - \hat{\lambda}^{k+1}\|^2
$$
$$
= \tfrac{\tau_k}{2\eta_k}\left[\|\hat{\lambda}^k - \lambda\|^2 - \|\hat{\lambda}^{k+1} - \lambda\|^2\right] + \tfrac{\eta_k \tau_k}{2}\|\tilde{s}^{k+1}\|^2. \qquad (28)
$$

Substituting (28) into (27) we obtain

$$
\begin{aligned}
\mathcal{L}_{\rho_k}(z^{k+1}, \lambda) \quad &\leq (1-\tau_k)\mathcal{L}_{\rho_{k-1}}(z^k, \lambda) + \tau_k F(z^\star) + \tfrac{\gamma_k \tau_k^2}{2}\left[\|\tilde{x}^k - x^\star\|^2 - \|\tilde{x}^{k+1} - x^\star\|^2\right] \\
&+ \tfrac{\beta_k \tau_k^2}{2}\left[\|\tilde{y}^k - y^\star\|^2 - \|\tilde{y}^{k+1} - y^\star\|^2\right] + \tfrac{\tau_k}{2\eta_k}\left[\|\hat{\lambda}^k - \lambda\|^2 - \|\hat{\lambda}^{k+1} - \lambda\|^2\right] \\
&+ \tfrac{\eta_k \tau_k}{2}\|\tilde{s}^{k+1}\|^2 - \tfrac{\rho_k \tau_k^2}{2}\|\tilde{s}^{k+1/2}\|^2 - \tfrac{(\beta_k - \rho_k L_B)\tau_k^2}{2}\|\tilde{y}^{k+1} - \tilde{y}^k\|^2 \\
&- \tfrac{(1-\tau_k)}{2}\left[\rho_{k-1} - \rho_k(1-\tau_k)\right]\|s^k\|^2.
\end{aligned}
\tag{29}
$$

Finally, by using $\|u\|^2 - 2\|v\|^2 \leq 2\|u - v\|^2$, it is straightforward to show that if $2\eta_k \leq \rho_k \tau_k$, then

$$
\tfrac{\eta_k \tau_k}{2}\|\tilde{s}^{k+1}\|^2 - \tfrac{\rho_k \tau_k^2}{2}\|\tilde{s}^{k+1/2}\|^2 \leq \tfrac{L_B \rho_k \tau_k^2}{2}\|\tilde{y}^{k+1} - \tilde{y}^k\|^2.
$$

Therefore, substituting this estimate into (29), we obtain (24). $\qquad\square$

From Lemma B.2, we need to derive rules for updating the parameters $\tau_k$, $\rho_k$, $\gamma_k$, $\beta_k$, and $\eta_k$. These updates are guided by the following lemma, which is shown in Algorithm 1.

**Lemma B.3.** *If the parameters $\tau_k$, $\rho_k$, $\gamma_k$, $\beta_k$, and $\eta_k$ are updated as*

$$
\begin{cases}
\tau_k := \frac{1}{k+1}, \quad \rho_k := \rho_0(k+1), \quad \beta_k := 2L_B \rho_0 (k+1), \\[2mm]
\eta_k := \frac{\rho_0}{2}, \quad \text{and} \quad 0 \leq \gamma_{k+1} \leq \left(\frac{k+2}{k+1}\right)\gamma_k,
\end{cases}
\tag{30}
$$

*then the sequence $\left\{(z^k, \tilde{z}^k)\right\}$ satisfies*

$$
2k S_{\rho_{k-1}}(z^k, \hat{\lambda}^0) + \tfrac{\gamma_k}{k+1}\|\tilde{x}^k - x^\star\|^2 + 2\rho_0 L_B\|\tilde{y}^k - y^\star\|^2 \leq \gamma_0\|x^0 - x^\star\|^2 + 2\rho_0 L_B\|y^0 - y^\star\|^2, \tag{31}
$$

*where $S_{\rho_{k-1}}(z^k, \hat{\lambda}^0) := \mathcal{L}_{\rho_{k-1}}(z^k, \hat{\lambda}^0) - F^\star$, and $\rho_0 > 0$ and $\gamma_0 \geq 0$ are given.*

*Proof.* First, we choose to update $\tau_k$ as $\tau_k = \frac{1}{k+1}$. Then, $\tau_0 = 1$. From the last term of (24), we impose $\rho_{k-1} - \rho_k(1-\tau_k) = 0$. This suggests us to update $\rho_k$ as $\rho_k = \rho_0(k+1)$.

We also choose $\beta_k := 2L_B\rho_k$ and $\eta_k := \frac{\rho_k \tau_k}{2}$ to guarantee $\beta_k - 2\rho_k L_B \geq 0$ and $2\eta_k \leq \rho_k \tau_k$, respectively. Using the update of $\tau_k$ and $\rho_k$, we can easily show that $\beta_k = 2L_B\rho_0(k+1)$ and $\eta_k := \frac{\rho_0}{2}$ as shown in (30).

Using the update (30) and $\lambda := \hat{\lambda}^0$ into (24) with $S_k := \mathcal{L}_{\rho_{k-1}}(z^k, \hat{\lambda}^0) - F^\star$, we have

$$
\begin{aligned}
(k+1)S_{k+1} + \tfrac{1}{\rho_0}\|\hat{\lambda}^{k+1} - \hat{\lambda}^0\|^2 &+ \tfrac{\gamma_k}{2(k+1)}\|\tilde{x}^{k+1} - x^\star\|^2 + \rho_0 L_B\|\tilde{y}^{k+1} - y^\star\|^2 \leq k S_k \\
&+ \tfrac{1}{\rho_0}\|\hat{\lambda}^k - \hat{\lambda}^0\|^2 + \tfrac{\gamma_k}{2(k+1)}\|\tilde{x}^k - x^\star\|^2 + \rho_0 L_B\|\tilde{y}^k - y^\star\|^2.
\end{aligned}
$$

We also choose $\frac{\gamma_{k+1}}{k+2} \leq \frac{\gamma_k}{k+1}$. Hence, by induction, the last inequality leads to

$$
k S_k + \tfrac{1}{\rho_0}\|\hat{\lambda}^k - \hat{\lambda}^0\|^2 + \tfrac{\gamma_k}{2(k+1)}\|\tilde{x}^k - x^\star\|^2 + \rho_0 L_B\|\tilde{y}^k - y^\star\|^2 \leq \tfrac{\gamma_0}{2}\|\tilde{x}^0 - x^\star\|^2 + \rho_0 L_B\|\tilde{y}^0 - y^\star\|^2.
$$

Since $\tilde{x}^0 = x^0$ and $\tilde{y}^0 = y^0$, by ignoring the term $\frac{1}{\rho_0}\|\hat{\lambda}^k - \hat{\lambda}^0\|^2$, the last inequality leads to (31). Finally, the condition $\frac{\gamma_{k+1}}{k+2} \leq \frac{\gamma_k}{k+1}$ holds if $0 \leq \gamma_{k+1} \leq \left(\frac{k+2}{k+1}\right)\gamma_k$. $\qquad\square$

***The proof of Theorem 3.1.*** Let $R_0^2 := \gamma_0\|x^0 - x^\star\|^2 + 2\rho_0 L_B\|y^0 - y^\star\|^2$. From (31), we have $S_{\rho_{k-1}}(z^k, \hat{\lambda}^0) = \mathcal{L}_{\rho_k}(z^k, \hat{\lambda}^0) - F^\star \leq \frac{R_0^2}{2k}$. Moreover, $\rho_{k-1} = \rho_0 k$. Substituting these two expressions into (6), we obtain (8). $\qquad\square$

## C  Lower bound on convergence rates of Algorithm 1

In order to show that the convergence rate of Algorithm 1 is optimal, we consider the following example studied in [28]:

$$
\min_{z:=[x,y]}\left\{F(z) := f(x) + g(y) \mid x - y = 0\right\}, \tag{32}
$$

which is a split reformulation of an additive composite objective function $F(x) = f(x) + g(x)$. Algorithm 1 for solving (32) can be cast as a special case of the following generic scheme:

$$\begin{cases} (\hat{y}^k, \hat{\lambda}^k) & \text{are linear combinations of previous iterates} \\ x^{k+1} & := \operatorname{prox}_{\gamma_k f}\big(\hat{x}^k - \gamma_k^{-1}\hat{\lambda}^k\big) \\ (\tilde{x}^{k+1}, \hat{\lambda}^{k+1}) & \text{are linear combinations of computed iterates} \\ y^{k+1} & := \operatorname{prox}_{\beta_k g}\big(\tilde{x}^{k+1} - \beta_k^{-1}\hat{\lambda}^{k+1}\big). \end{cases} \quad (33)$$

Then, there exist $f$ and $g$ defined on $\big\{x \in \mathbb{R}^{6k+5} \mid \|x\| \le B\big\}$ which are convex and $L_f$-Lipschitz continuous such that the general primal-dual scheme (33) exhibits a lower bound:

$$F(\breve{x}^k) \ge \frac{L_f B}{8(k+1)},$$

where $\breve{x}^k := \sum_{j=1}^k \alpha_j x^j + \sum_{l=1}^k \sigma_l y^l$ for any $\alpha_j$ and $\sigma_l$ with $j,l = 1, \cdots, k$. This example can be found in [14, Proposition 5]. Consequently, Algorithm 1 has a lower bound convergence rate of $\mathcal{O}\big(\frac{1}{k}\big)$. Hence, the $\mathcal{O}\big(\frac{1}{k}\big)$ convergence rate stated in Theorem 3.1 is optimal within a constant factor.

# D  Convergence analysis of Algorithm 2

Lemmas D.1 and D.2 provide key estimates to prove the convergence of Algorithm 2.

**Lemma D.1.** *Assume that $\mathcal{L}_\rho$ is defined by (5), and $\ell_\rho^k$ is defined by (14). Let $\mathcal{Q}_\rho^k$ be defined as*

$$\mathcal{Q}_{\rho_k}^k(y) := \phi_{\rho_k}(\hat{z}^{k+1}, \hat{\lambda}^k) + \langle \nabla_y \phi_{\rho_k}(\hat{z}^{k+1}, \hat{\lambda}^k), y - \hat{y}^k \rangle + \frac{\rho_k L_B}{2}\|y - \hat{y}^k\|^2. \quad (34)$$

*Then, $\phi_{\rho_k}(x^{k+1}, y, \hat{\lambda}^k) \le \mathcal{Q}_{\rho_k}^k(y)$ for any $y \in \mathbb{R}^{\hat{p}}$.*

*Let $(x^{k+1}, \hat{z}^{k+1}, \hat{z}^k, \hat{\lambda}^k)$ be computed by Algorithm 2, and $\breve{y}^{k+1} := (1 - \tau_k)y^k + \tau_k \tilde{y}^{k+1}$. Then, for any $z \in \operatorname{dom}(F)$, we have*

$$\begin{aligned} \breve{\mathcal{L}}_{\rho_k}^{k+1} &:= f(x^{k+1}) + g(\breve{y}^{k+1}) + \mathcal{Q}_{\rho_k}^k(\breve{y}^{k+1}) \le (1 - \tau_k)\big[F(z^k) + \ell_{\rho_k}^k(z^k)\big] \\ &+ \tau_k\big[F(z) + \ell_{\rho_k}^k(z)\big] + \frac{\gamma_k \tau_k^2}{2}\|\tilde{x}^k - x\|^2 - \frac{\gamma_k \tau_k^2}{2}\|\tilde{x}^{k+1} - x\|^2 - \frac{\gamma_k}{2}\|x^{k+1} - \hat{x}^k\|^2 \\ &+ \frac{\beta_k \tau_k^2}{2}\|\tilde{y}^k - y\|^2 - \frac{\beta_k \tau_k^2 + \mu_g \tau_k}{2}\|\tilde{y}^{k+1} - y\|^2 - \frac{(\beta_k - \rho_k L_B)\tau_k^2}{2}\|\tilde{y}^{k+1} - \tilde{y}^k\|^2. \end{aligned} \quad (35)$$

*Proof.* Since $\hat{z}^k = (1 - \tau_k)z^k + \tau_k \hat{z}^k$, we have $(1 - \tau_k)x^k + \tau_k \tilde{x}^{k+1} - x^{k+1} = 0$ and $\breve{y}^{k+1} - \hat{y}^k = \tau_k(\tilde{y}^{k+1} - \tilde{y}^k)$. Using these expressions, $\breve{y}^{k+1}$, $\ell_{\rho_k}^k$ in (14), and $\mathcal{Q}_{\rho_k}^k$ in (34), we can derive

$$\begin{aligned} \mathcal{Q}_{\rho_k}^k(\breve{y}^{k+1}) &= \phi_{\rho_k}(\hat{z}^{k+1}, \hat{\lambda}^k) + \langle \nabla_y \phi_{\rho_k}(\hat{z}^{k+1}, \hat{\lambda}^k), \breve{y}^{k+1} - \hat{y}^k \rangle + \frac{\rho_k L_B}{2}\|\breve{y}^{k+1} - \hat{y}^k\|^2 \\ &= (1 - \tau_k)\Big[\phi_{\rho_k}(\hat{z}^{k+1}, \hat{\lambda}^k) + \langle \nabla_x \phi_{\rho_k}(\hat{z}^{k+1}, \hat{\lambda}^k), x^k - x^{k+1} \rangle + \langle \nabla_y \phi_{\rho_k}(\hat{z}^{k+1}, \hat{\lambda}^k), y^k - \hat{y}^k \rangle\Big] \\ &+ \tau_k\Big[\phi_{\rho_k}(\hat{z}^{k+1}, \hat{\lambda}^k) + \langle \nabla_x \phi_{\rho_k}(\hat{z}^{k+1}, \hat{\lambda}^k), \tilde{x}^{k+1} - x^{k+1} \rangle + \langle \nabla_y \phi_{\rho_k}(\hat{z}^{k+1}, \hat{\lambda}^k), \tilde{y}^{k+1} - \hat{y}^k \rangle\Big] \\ &+ \langle \nabla_x \phi_{\rho_k}(\hat{z}^{k+1}, \hat{\lambda}^k), (1 - \tau_k)x^k + \tau_k \tilde{x}^{k+1} - x^{k+1} \rangle + \frac{\rho_k \tau_k^2 L_B}{2}\|\tilde{y}^{k+1} - \tilde{y}^k\|^2 \\ &\overset{(14)}{=} (1 - \tau_k)\ell_{\rho_k}^k(z^k) + \tau_k \ell_{\rho_k}^k(\hat{z}^{k+1}) + \frac{\rho_k \tau_k^2 L_B}{2}\|\tilde{y}^{k+1} - \tilde{y}^k\|^2. \end{aligned} \quad (36)$$

By the convexity of $f$ and $x^{k+1} - (1 - \tau_k)x^k = \tau_k \tilde{x}^{k+1}$, for any $x \in \operatorname{dom}(f)$ and $\nabla f(x^{k+1}) \in \partial f(x^{k+1})$, we can estimate that

$$\begin{aligned} f(x^{k+1}) &\le f((1 - \tau_k)x^k + \tau_k x) + \langle \nabla f(x^{k+1}), x^{k+1} - (1 - \tau_k)x^k - \tau_k x \rangle \\ &\le (1 - \tau_k)f(x^k) + \tau_k f(x) + \tau_k \langle \nabla f(x^{k+1}), \tilde{x}^{k+1} - x \rangle, \end{aligned} \quad (37)$$

Since $\breve{y}^{k+1} := (1 - \tau_k)y^k + \tau_k \tilde{y}^{k+1}$, by $\mu_g$-convexity of $g$, for any $y \in \operatorname{dom}(g)$ and $\nabla g(\tilde{y}^{k+1}) \in \partial g(\tilde{y}^{k+1})$, we have

$$\begin{aligned} g(\breve{y}^{k+1}) &\le (1 - \tau_k)g(y^k) + \tau_k g(\tilde{y}^{k+1}) - \frac{\tau_k(1 - \tau_k)\mu_g}{2}\|\tilde{y}^{k+1} - y^k\|^2 \\ &\le (1 - \tau_k)g(y^k) + \tau_k g(y) + \tau_k \langle \nabla g(\tilde{y}^{k+1}), \tilde{y}^{k+1} - y \rangle - \frac{\tau_k \mu_g}{2}\|\tilde{y}^{k+1} - y\|^2. \end{aligned} \quad (38)$$

Next, note that

$$\ell_{\rho_k}^k(\check{z}^{k+1}) = \phi_{\rho_k}(\hat{z}^{k+1}, \hat{\lambda}^k) + \langle \nabla_x \phi_{\rho_k}(\hat{z}^{k+1}, \hat{\lambda}^k), \tilde{x}^{k+1} - x^{k+1} \rangle + \langle \nabla_y \phi_{\rho_k}(\hat{z}^{k+1}, \hat{\lambda}^k), \tilde{y}^{k+1} - \hat{y}^k \rangle$$

$$= \phi_{\rho_k}(\hat{z}^{k+1}, \hat{\lambda}^k) + \langle \nabla_x \phi_{\rho_k}(\hat{z}^{k+1}, \hat{\lambda}^k), x - x^{k+1} \rangle + \langle \nabla_y \phi_{\rho_k}(\hat{z}^{k+1}, \hat{\lambda}^k), y - \hat{y}^k \rangle \tag{39}$$

$$+ \langle \nabla_x \phi_{\rho_k}(\hat{z}^{k+1}, \hat{\lambda}^k), \tilde{x}^{k+1} - x \rangle + \langle \nabla_y \phi_{\rho_k}(\hat{z}^{k+1}, \hat{\lambda}^k), \tilde{y}^{k+1} - y \rangle$$

$$= \ell_{\rho_k}^k(z) + \langle \nabla_x \phi_{\rho_k}(\hat{z}^{k+1}, \hat{\lambda}^k), \tilde{x}^{k+1} - x \rangle + \langle \nabla_y \phi_{\rho_k}(\hat{z}^{k+1}, \hat{\lambda}^k), \tilde{y}^{k+1} - y \rangle.$$

Combining (36), (37), (38), and (39), for any $z := (x, y) \in \operatorname{dom}(F)$, we can derive

$$\check{\mathcal{L}}_{\rho_k}^{k+1} \overset{(34)}{=} f(x^{k+1}) + g(\check{y}^{k+1}) + \mathcal{Q}_{\rho_k}^k(\check{y}^{k+1})$$

$$\overset{(37),(38),(39)}{\leq} (1 - \tau_k) \left[ F(z^k) + \ell_{\rho_k}^k(z^k) \right] + \tau_k \left[ F(z) + \ell_{\rho_k}^k(z) \right] \tag{40}$$

$$+ \tau_k \langle \nabla f(x^{k+1}) + \nabla_x \phi_{\rho_k}(\hat{z}^{k+1}, \hat{\lambda}^k), \tilde{x}^{k+1} - x \rangle + \tau_k \langle \nabla g(\tilde{y}^{k+1}) + \nabla_y \phi_{\rho_k}(\hat{z}^{k+1}, \hat{\lambda}^k), \tilde{y}^{k+1} - y \rangle$$

$$- \frac{\tau_k \mu_g}{2} \|\tilde{y}^{k+1} - y\|^2 + \frac{\rho_k \tau_k^2 L_B}{2} \|\tilde{y}^{k+1} - \tilde{y}^k\|^2.$$

Next, from the optimality condition of the $x$- and $y_i$-subproblems in Algorithm 2, we can show that

$$\begin{cases} \nabla f(x^{k+1}) + \nabla_x \phi_{\rho_k}(\hat{z}^{k+1}, \hat{\lambda}^k) = \gamma_k(\hat{x}^k - x^{k+1}), & \nabla f(x^{k+1}) \in \partial f(x^{k+1}), \\ \nabla g(\tilde{y}^{k+1}) + \nabla_y \phi_{\rho_k}(\hat{z}^{k+1}, \hat{\lambda}^k) = \tau_k \beta_k(\tilde{y}^k - \tilde{y}^{k+1}), & \nabla g(\tilde{y}^{k+1}) \in \partial g(\tilde{y}^{k+1}). \end{cases} \tag{41}$$

Moreover, we also have

$$\begin{aligned} 2\tau_k \langle \hat{x}^k - x^{k+1}, \tilde{x}^{k+1} - x \rangle &= \tau_k^2 \|\tilde{x}^k - x\|^2 - \tau_k^2 \|\tilde{x}^{k+1} - x\|^2 - \|x^{k+1} - \hat{x}^k\|^2 \\ 2\langle \tilde{y}^k - \tilde{y}^{k+1}, \tilde{y}^{k+1} - y \rangle &= \|\tilde{y}^k - y\|^2 - \|\tilde{y}^{k+1} - y\|^2 - \|\tilde{y}^{k+1} - \tilde{y}^k\|^2. \end{aligned} \tag{42}$$

Using (41) and (42) into (40), we can further derive

$$\check{\mathcal{L}}_{\rho_k}^{k+1} \overset{(35)}{\leq} (1 - \tau_k) \left[ F(z^k) + \ell_{\rho_k}^k(z^k) \right] + \tau_k \left[ F(z) + \ell_{\rho_k}^k(z) \right] - \frac{\tau_k \mu_g}{2} \|\tilde{y}^{k+1} - y\|^2$$

$$+ \tau_k \gamma_k \langle \hat{x}^k - x^{k+1}, \tilde{x}^{k+1} - x \rangle + \tau_k^2 \beta_k \langle \tilde{y}^k - \tilde{y}^{k+1}, \tilde{y}^{k+1} - y \rangle + \frac{\rho_k \tau_k^2 L_B}{2} \|\tilde{y}^{k+1} - \tilde{y}^k\|^2$$

$$\overset{(42)}{\leq} (1 - \tau_k) \left[ F(z^k) + \ell_{\rho_k}^k(z^k) \right] + \tau_k \left[ F(z) + \ell_{\rho_k}^k(z) \right]$$

$$+ \frac{\gamma_k \tau_k^2}{2} \|\tilde{x}^k - x\|^2 - \frac{\gamma_k \tau_k^2}{2} \|\tilde{x}^{k+1} - x\|^2 - \frac{\gamma_k}{2} \|x^{k+1} - \hat{x}^k\|^2$$

$$+ \frac{\beta_k \tau_k^2}{2} \|\tilde{y}^k - y\|^2 - \frac{(\beta_k \tau_k^2 + \mu_g \tau_k)}{2} \|\tilde{y}^{k+1} - y\|^2 - \frac{(\beta_k - \rho_k L_B)\tau_k^2}{2} \|\tilde{y}^{k+1} - \tilde{y}^k\|^2,$$

which is exactly (35). $\qquad\square$

**Lemma D.2.** *Let* $\{(z^k, \hat{z}^k, \tilde{z}^k, \hat{\lambda}^k)\}$ *be the sequence generated by Algorithm 2. Then*

$$\mathcal{L}_{\rho_k}(z^{k+1}, \hat{\lambda}^k) \leq (1 - \tau_k)\mathcal{L}_{\rho_{k-1}}(z^k, \hat{\lambda}^k) + \tau_k F(z^\star) - \frac{(1 - \tau_k)}{2}(\rho_{k-1} - \rho_k(1 - \tau_k))\|s^k\|^2$$

$$+ \frac{\gamma_k \tau_k^2}{2} \|\tilde{x}^k - x^\star\|^2 - \frac{\gamma_k \tau_k^2}{2} \|\tilde{x}^{k+1} - x^\star\|^2 - \frac{\gamma_k}{2} \|x^{k+1} - \hat{x}^k\|^2 \tag{43}$$

$$+ \frac{\beta_k \tau_k^2}{2} \|\tilde{y}^k - y^\star\|^2 - \frac{(\beta_k \tau_k^2 + \mu_g \tau_k)}{2} \|\tilde{y}^{k+1} - y^\star\|^2 - \frac{(\beta_k - \rho_k L_B)\tau_k^2}{2} \|\tilde{y}^{k+1} - \tilde{y}^k\|^2$$

$$- \langle \hat{\lambda}^k - \hat{\lambda}^0, B(y^{k+1} - \check{y}^{k+1}) \rangle - \frac{\rho_k L_B}{2} \|y^{k+1} - \check{y}^{k+1}\|^2 - \frac{\rho_k \tau_k^2}{2} \|\tilde{s}^{k+1/2}\|^2,$$

*where* $\gamma_k$, $\beta_k$, *and* $\rho_k$ *are positive parameters,* $\tau_k \in [0, 1]$, $s^k := Ax^k + By^k - c$, $\tilde{s}^{k+1/2} := A\tilde{x}^{k+1} + B\tilde{x}^k - c$, *and* $\check{y}^{k+1} := (1 - \tau_k)y^k + \tau_k \tilde{y}^{k+1}$.

*Proof.* Using (35) with $z = z^\star$, and then combining the result with (15), we obtain

$$\check{\mathcal{L}}_{\rho_k}^{k+1} \leq (1 - \tau_k)\mathcal{L}_{\rho_k}(z^k, \hat{\lambda}^k) + \tau_k F(z^\star) - \frac{(1 - \tau_k)\rho_k}{2} \|\hat{s}^{k+1} - s^k\|^2 - \frac{\rho_k \tau_k}{2} \|\hat{s}^{k+1}\|^2$$

$$+ \frac{\gamma_k \tau_k^2}{2} \|\tilde{x}^k - x^\star\|^2 - \frac{\gamma_k \tau_k^2}{2} \|\tilde{x}^{k+1} - x^\star\|^2 - \frac{\gamma_k}{2} \|x^{k+1} - \hat{x}^k\|^2$$

$$+ \frac{\beta_k \tau_k^2}{2} \|\tilde{y}^k - y\|^2 - \frac{(\beta_k \tau_k^2 + \mu_g \tau_k)}{2} \|\tilde{y}^{k+1} - y\|^2 - \frac{(\beta_k - \rho_k L_B)\tau_k^2}{2} \|\tilde{y}^{k+1} - \tilde{y}^k\|^2.$$

Next, using $\mathcal{L}_{\rho_k}(z^k, \hat{\lambda}^k) = \mathcal{L}_{\rho_{k-1}}(z^k, \hat{\lambda}^k) + \frac{(\rho_k - \rho_{k-1})}{2}\|s^k\|^2$ in the last inequality, and then combining the result with (26), we obtain

$$
\begin{aligned}
\check{\mathcal{L}}_{\rho_k}^{k+1} \leq{} & (1-\tau_k)\mathcal{L}_{\rho_{k-1}}(z^k, \hat{\lambda}^k) + \tau_k F(z^\star) - \frac{(1-\tau_k)(\rho_{k-1}-\rho_k(1-\tau_k))}{2}\|s^k\|^2 - \frac{\rho_k \tau_k^2}{2}\|\tilde{s}^{k+1/2}\|^2 \\
& + \frac{\gamma_k \tau_k^2}{2}\|\tilde{x}^k - x^\star\|^2 - \frac{\gamma_k \tau_k^2}{2}\|\tilde{x}^{k+1} - x^\star\|^2 - \frac{\gamma_k}{2}\|x^{k+1} - \hat{x}^k\|^2 \\
& + \frac{\beta_k \tau_k^2}{2}\|\tilde{y}^k - y^\star\|^2 - \frac{(\beta_k \tau_k^2 + \mu_g \tau_k)}{2}\|\tilde{y}^{k+1} - y^\star\|^2 - \frac{(\beta_k - \rho_k L_B)\tau_k^2}{2}\|\tilde{y}^{k+1} - \tilde{y}^k\|^2.
\end{aligned}
\tag{44}
$$

Now, we consider two cases corresponding to the two options at Step 11 of Algorithm 2.

**Option 1:** If $y^{k+1} = \breve{y}^{k+1}$, then we have

$$
\begin{aligned}
\mathcal{L}_{\rho_k}(z^{k+1}, \hat{\lambda}^k) ={} & f(x^{k+1}) + g(y^{k+1}) + \phi_{\rho_k}(z^{k+1}, \hat{\lambda}^k) \\
\overset{(17)}{\leq}{} & f(x^{k+1}) + g(\breve{y}^{k+1}) + \phi_{\rho_k}(\hat{z}^{k+1}, \hat{\lambda}^k) + \langle \nabla_y \phi_{\rho_k}(\hat{z}^{k+1}, \hat{\lambda}^k), \breve{y}^{k+1} - \hat{y}^k \rangle \\
& + \frac{\rho_k L_B}{2}\|\breve{y}^{k+1} - \hat{y}^k\|^2 \\
={} & f(x^{k+1}) + g(\breve{y}^{k+1}) + \mathcal{Q}_{\rho_k}^k(\breve{y}^{k+1}) \\
={} & \check{\mathcal{L}}_{\rho_k}^{k+1} \\
={} & \check{\mathcal{L}}_{\rho_k}^{k+1} - \langle \hat{\lambda}^k - \hat{\lambda}^0, B(y^{k+1} - \breve{y}^{k+1}) \rangle - \frac{\rho_k L_B}{2}\|y^{k+1} - \breve{y}^{k+1}\|^2.
\end{aligned}
$$

Here, the last relation follows from the fact that $\langle \hat{\lambda}^k - \hat{\lambda}^0, B(y^{k+1} - \breve{y}^{k+1}) \rangle + \frac{\rho_k L_B}{2}\|y^{k+1} - \breve{y}^{k+1}\|^2 = 0$ since $y^{k+1} = \breve{y}^{k+1}$. Combining the last estimate and (44), we obtain the key estimate (43).

**Option 2:** If we choose $y_i^{k+1} := \operatorname{prox}_{g_i/(\rho_k L_B)}\big(\hat{y}_i^k - \frac{1}{\rho_k L_B} B_i^\top(\rho_k r^k - \hat{\lambda}^0)\big)$, then we write it as

$$
y_i^{k+1} = \underset{y_i}{\arg\min}\left\{ g_i(y_i) + \langle \nabla_{y_i}\phi_{\rho_k}(\hat{z}^{k+1}, \hat{\lambda}^0), y_i - \hat{y}_i^k \rangle + \frac{\rho_k L_B}{2}\|y_i - \hat{y}_i^k\|^2 \right\} \quad \text{for all } i = 1, \cdots, m.
$$

From the optimality condition of these $y_i$-subproblems, one can easily show that

$$
\begin{aligned}
& g(y^{k+1}) + \langle \nabla_y \phi_{\rho_k}(\hat{z}^{k+1}, \hat{\lambda}^0), y^{k+1} - \hat{y}^k \rangle + \frac{\rho_k L_B}{2}\|y^{k+1} - \hat{y}^k\|^2 \\
& \leq g(\breve{y}^{k+1}) + \langle \nabla_y \phi_{\rho_k}(\hat{z}^{k+1}, \hat{\lambda}^0), \breve{y}^{k+1} - \hat{y}^k \rangle + \frac{\rho_k L_B}{2}\|\breve{y}^{k+1} - \hat{y}^k\|^2 - \frac{\rho_k L_B}{2}\|y^{k+1} - \breve{y}^{k+1}\|^2.
\end{aligned}
$$

Using $\phi_{\rho_k}(x^{k+1}, \breve{y}^{k+1}, \hat{\lambda}^k) \leq \mathcal{Q}_{\rho_k}^k(\breve{y}^{k+1})$ from Lemma D.1, and the last inequality, we can derive

$$
\begin{aligned}
\mathcal{L}_{\rho_k}(z^{k+1}, \hat{\lambda}^k) ={} & f(x^{k+1}) + g(y^{k+1}) + \phi_{\rho_k}(z^{k+1}, \hat{\lambda}^k) \\
\overset{(17)}{\leq}{} & f(x^{k+1}) + g(y^{k+1}) + \phi_{\rho_k}(\hat{z}^{k+1}, \hat{\lambda}^k) + \langle \nabla_y \phi_{\rho_k}(\hat{z}^{k+1}, \hat{\lambda}^k), y^{k+1} - \hat{y}^k \rangle \\
& + \frac{\rho_k L_B}{2}\|y^{k+1} - \hat{y}^k\|^2 \\
={} & f(x^{k+1}) + \phi_{\rho_k}(\hat{z}^{k+1}, \hat{\lambda}^k) - \langle B^\top(\hat{\lambda}^k - \hat{\lambda}^0), y^{k+1} - \hat{y}^k \rangle \\
& + g(y^{k+1}) + \langle \nabla_y \phi_{\rho_k}(\hat{z}^{k+1}, \hat{\lambda}^0), y^{k+1} - \hat{y}^k \rangle + \frac{\rho_k L_B}{2}\|y^{k+1} - \hat{y}^k\|^2 \\
\leq{} & f(x^{k+1}) + \phi_{\rho_k}(\hat{z}^{k+1}, \hat{\lambda}^k) - \langle B^\top(\hat{\lambda}^k - \hat{\lambda}^0), y^{k+1} - \hat{y}^k \rangle - \frac{\rho_k L_B}{2}\|y^{k+1} - \breve{y}^{k+1}\|^2 \\
& + g(\breve{y}^{k+1}) + \langle \nabla_y \phi_{\rho_k}(\hat{z}^{k+1}, \hat{\lambda}^0), \breve{y}^{k+1} - \hat{y}^k \rangle + \frac{\rho_k L_B}{2}\|\breve{y}^{k+1} - \hat{y}^k\|^2 \\
\leq{} & \check{\mathcal{L}}_{\rho_k}^{k+1} - \frac{\rho_k L_B}{2}\|y^{k+1} - \breve{y}^{k+1}\|^2 - \langle \hat{\lambda}^k - \hat{\lambda}^0, B(y^{k+1} - \breve{y}^{k+1}) \rangle.
\end{aligned}
$$

Combining this estimate and (44), we obtain the key estimate (43). $\square$

Our next step is to show how to choose the parameters $\gamma_k, \beta_k, \rho_k$, and $\tau_k \in [0,1]$ such that we can obtain a convergence property of $\mathcal{L}_{\rho_k}(\cdot)$.

**Lemma D.3.** *If the parameters $\tau_k, \rho_k, \gamma_k, \beta_k$, and $\eta_k$ are updated as*

$$
\begin{cases}
\tau_k := \frac{1}{2}\tau_{k-1}\big((\tau_{k-1}^2 + 4)^{1/2} - \tau_{k-1}\big), & \rho_k := \frac{\rho_0}{\tau_k^2}, \\
\gamma_k := \gamma_0 \geq 0, & \beta_k := 2L_B \rho_k, \quad \text{and} \quad \eta_k := \frac{\rho_k \tau_k}{2},
\end{cases}
\tag{45}
$$

*with $\tau_0 := 1$ and $\rho_0 \in \left(0, \frac{\mu_g}{4L_B}\right]$, then*

$$
\mathcal{L}_{\rho_{k-1}}(z^k, \hat{\lambda}^0) - F(z^\star) \leq \frac{\tau_{k-1}^2}{2}\left[\gamma_0\|\tilde{x}^0 - x^\star\|^2 + 2\rho_0 L_B\|\tilde{y}^0 - y^\star\|^2\right].
\tag{46}
$$

*Proof.* Since $\mathcal{L}_\rho(z, \hat\lambda^0) = \mathcal{L}_\rho(z, \hat\lambda^k) + \langle \hat\lambda^k - \hat\lambda^0, Ax + By - c \rangle$, from (43), we have

$$
\begin{aligned}
\mathcal{L}_{\rho_k}(z^{k+1}, \hat\lambda^0) &\leq (1-\tau_k)\mathcal{L}_{\rho_{k-1}}(z^k, \hat\lambda^0) + \tau_k F(z^\star) - \tfrac{(1-\tau_k)}{2}(\rho_{k-1} - \rho_k(1-\tau_k))\|s^k\|^2 \\
&\quad + \tfrac{\gamma_k \tau_k^2}{2}\|\tilde x^k - x^\star\|^2 - \tfrac{\gamma_k \tau_k^2}{2}\|\tilde x^{k+1} - x^\star\|^2 - \tfrac{\gamma_k}{2}\|x^{k+1} - \hat x^k\|^2 \\
&\quad + \tfrac{\beta_k \tau_k^2}{2}\|\tilde y^k - y^\star\|^2 - \tfrac{(\beta_k \tau_k^2 + \mu_g \tau_k)}{2}\|\tilde y^{k+1} - y^\star\|^2 - \tfrac{(\beta_k - \rho_k L_B)\tau_k^2}{2}\|\tilde y^{k+1} - \tilde y^k\|^2 \quad (47) \\
&\quad + \langle \hat\lambda^k - \hat\lambda^0, Ax^{k+1} + By^{k+1} - c - (1-\tau_k)(Ax^k + By^k - c)\rangle \\
&\quad - \langle \hat\lambda^k - \hat\lambda^0, B(y^{k+1} - \breve y^{k+1})\rangle - \tfrac{\rho_k L_B}{2}\|y^{k+1} - \breve y^{k+1}\|^2 - \tfrac{\rho_k \tau_k^2}{2}\|\tilde s^{k+1/2}\|^2.
\end{aligned}
$$

Now, using $\breve y^{k+1} - (1-\tau_k)y^k = \tau_k \tilde y^{k+1}$, $x^{k+1} - (1-\tau_k)x^k = \tau_k \tilde x^{k+1}$, and the dual update $\hat\lambda^{k+1} := \hat\lambda^k - \eta_k(A\tilde x^{k+1} + B\tilde y^{k+1} - c) = \hat\lambda^k - \eta_k \tilde s^{k+1}$, we can show that

$$
\begin{aligned}
M_k &:= \langle \hat\lambda^k - \hat\lambda^0, Ax^{k+1} + By^{k+1} - c - (1-\tau_k)(Ax^k + By^k - c) - B(y^{k+1} - \breve y^{k+1})\rangle \\
&= \langle \hat\lambda^k - \hat\lambda^0, Ax^{k+1} + B\breve y^{k+1} - c - (1-\tau_k)(Ax^k + By^k - c)\rangle \\
&= \tau_k \langle \hat\lambda^k - \hat\lambda^0, A\tilde x^{k+1} + B\tilde y^{k+1} - c\rangle \\
&= \tfrac{\tau_k}{\eta_k}\langle \hat\lambda^k - \hat\lambda^0, \hat\lambda^k - \hat\lambda^{k+1}\rangle = \tfrac{\tau_k}{2\eta_k}\big[\|\hat\lambda^k - \hat\lambda^0\|^2 - \|\hat\lambda^{k+1} - \hat\lambda^0\|^2\big] + \tfrac{\eta_k \tau_k}{2}\|\tilde s^{k+1}\|^2.
\end{aligned}
$$

Using this estimate of $M_k$ into (47), similar to (29), if $2\eta_k \leq \rho_k \tau_k$, then we can show that

$$
\begin{aligned}
\mathcal{L}_{\rho_k}(z^{k+1}, \hat\lambda^0) &\leq (1-\tau_k)\mathcal{L}_{\rho_{k-1}}(z^k, \hat\lambda^0) + \tau_k F(z^\star) - \tfrac{(1-\tau_k)}{2}(\rho_{k-1} - \rho_k(1-\tau_k))\|s^k\|^2 \\
&\quad + \tfrac{\gamma_k \tau_k^2}{2}\|\tilde x^k - x^\star\|^2 - \tfrac{\gamma_k \tau_k^2}{2}\|\tilde x^{k+1} - x^\star\|^2 + \tfrac{\beta_k \tau_k^2}{2}\|\tilde y^k - y^\star\|^2 \\
&\quad - \tfrac{(\beta_k \tau_k^2 + \mu_g \tau_k)}{2}\|\tilde y^{k+1} - y^\star\|^2 - \tfrac{(\beta_k - 2\rho_k L_B)\tau_k^2}{2}\|\tilde y^{k+1} - \tilde y^k\|^2 \quad (48) \\
&\quad - \tfrac{\rho_k L_B}{2}\|y^{k+1} - \breve y^{k+1}\|^2 + \tfrac{\tau_k}{2\eta_k}\big[\|\hat\lambda^k - \hat\lambda^0\|^2 - \|\hat\lambda^{k+1} - \hat\lambda^0\|^2\big].
\end{aligned}
$$

Let us first update $\tau_k$ as $\tau_k = \tfrac{1}{2}\tau_{k-1}\big((\tau_{k-1}^2 + 4)^{1/2} - \tau_{k-1}\big)$ with $\tau_0 = 1$, and $\rho_k = \tfrac{\rho_{k-1}}{1-\tau_k}$ as in (45). It is not hard to show that $\tfrac{1}{k+1} \leq \tau_k \leq \tfrac{2}{k+2}$ and $\rho_k = \tfrac{\rho_0}{\tau_k^2}$. Moreover, $\prod_{i=1}^{k-1}(1-\tau_i) = \tfrac{1}{\tau_{k-1}^2} \leq \tfrac{4}{(k+1)^2}$.
To guarantee $\beta_k \geq 2L_B \rho_k$ and $2\eta_k \leq \rho_k \tau_k$, we can update $\beta_k := 2L_B \rho_k$ and $\eta_k := \tfrac{\rho_k \tau_k}{2}$. Therefore, (48) can be simplified as

$$
\begin{aligned}
\mathcal{L}_{\rho_k}(z^{k+1}, \hat\lambda^0) &\leq (1-\tau_k)\mathcal{L}_{\rho_{k-1}}(z^k, \hat\lambda^0) + \tau_k F(z^\star) + \tfrac{\gamma_k \tau_k^2}{2}\|\tilde x^k - x^\star\|^2 \\
&\quad - \tfrac{\gamma_k \tau_k^2}{2}\|\tilde x^{k+1} - x^\star\|^2 + \tfrac{\beta_k \tau_k^2}{2}\|\tilde y^k - y^\star\|^2 - \tfrac{(\beta_k \tau_k^2 + \mu_g \tau_k)}{2}\|\tilde y^{k+1} - y^\star\|^2 \quad (49) \\
&\quad + \tfrac{1}{\rho_k}\big[\|\hat\lambda^k - \hat\lambda^0\|^2 - \|\hat\lambda^{k+1} - \hat\lambda^0\|^2\big].
\end{aligned}
$$

Now, let us define

$$
A_k := \mathcal{L}_{\rho_{k-1}}(z^k, \hat\lambda^0) - F^\star + \tfrac{1}{\rho_k}\|\hat\lambda^k - \hat\lambda^0\|^2 + \tfrac{\gamma_{k-1}\tau_{k-1}^2}{2}\|\tilde x^k - x^\star\|^2 + \tfrac{(\beta_{k-1}\tau_{k-1}^2 + \mu_g \tau_{k-1})}{2}\|\tilde y^k - y^\star\|^2.
$$

Assume that

$$
\tfrac{1}{\rho_k} \leq \tfrac{1}{\rho_{k-1}}, \qquad \beta_k \tau_k^2 \leq \beta_{k-1}\tau_{k-1}^2 + \mu_g \tau_{k-1} \quad \text{and} \quad \gamma_k \tau_k^2 \leq \gamma_{k-1}\tau_{k-1}^2. \qquad (50)
$$

Then, (49) implies $A_{k+1} \leq (1-\tau_k)A_k$. By induction, and $\tau_0 = 1$, we can show that

$$
A_k \leq \tfrac{1}{2}\left(\prod_{i=1}^{k-1}(1-\tau_i)\right)\big[\gamma_0 \|\tilde x^0 - x^\star\|^2 + \beta_0 \|\tilde y^0 - y^\star\|^2\big],
$$

Since $\prod_{i=1}^{k-1}(1-\tau_i) = \tau_{k-1}^2$ and $\beta_0 = 2L_B \rho_0$, the last inequality implies $S_{\rho_{k-1}}(z^k, \hat\lambda^0) := \mathcal{L}_{\rho_{k-1}}(z^k, \hat\lambda^0) - F(z^\star) \leq \tfrac{\tau_{k-1}^2}{2}\big[\gamma_0 \|\tilde x^0 - x^\star\|^2 + 2\rho_0 L_B \|\tilde y^0 - y^\star\|^2\big]$, which proves (46).

Since $\beta_k := 2L_B \rho_k$, the condition $\tfrac{\beta_k \tau_k^2}{1-\tau_k} \leq \beta_{k-1}\tau_{k-1}^2 + \mu_g \tau_{k-1}$ becomes $L_B \rho_k \tfrac{\tau_k^2}{1-\tau_k} \leq L_B \rho_{k-1}\tau_{k-1}^2 + \tfrac{\mu_g}{2}\tau_{k-1}$. Using $\rho_k = \tfrac{\rho_0}{\tau_k^2}$ and $\tfrac{\tau_k^2}{1-\tau_k} = \tau_{k-1}^2$, the last condition holds if $L_B \rho_0 \tfrac{\tau_{k-1}}{\tau_k} \leq \tfrac{\mu_g}{2}$. Since $1 \leq \tfrac{\tau_{k-1}}{\tau_k} \leq 2$, $L_B \rho_0 \tfrac{\tau_{k-1}}{\tau_k} \leq \tfrac{\mu_g}{2}$ holds if $4L_B \rho_0 \leq \mu_g$. This condition leads to $\rho_0 \leq \tfrac{\mu_g}{4L_B}$.

Next, the condition $\frac{\gamma_k \tau_k^2}{1-\tau_k} \leq \gamma_{k-1} \tau_{k-1}^2$ shows that we can choose $\gamma_k$ as $\gamma_k \leq \gamma_{k-1}$. This condition holds if we fix $\gamma_k := \gamma_0 \geq 0$. Now, we find the condition for $\eta_k$ in (45). Since $\rho_k = \frac{\rho_0}{\tau_k^2}$, the condition $\frac{1}{\rho_k} \leq \frac{1}{\rho_{k-1}}$ in (50) is automatically satisfied. $\qquad\square$

***The proof of Theorem*** *3.2.* Let $R_0^2 := \gamma_0 \|x^0 - x^\star\|^2 + 2\rho_0 L_B \|y^0 - y^\star\|^2$. Since $\tilde{x}^0 = x^0$ and $\tilde{y}^0 = y^0$, from (46), we have $S_{\rho_{k-1}}(z^k, \hat{\lambda}^0) = \mathcal{L}_{\rho_{k-1}}(z^k, \hat{\lambda}^0) - F^\star \leq \tau_{k-1}^2 R_0^2 \leq \frac{2R_0^2}{(k+1)^2}$. Moreover, $\rho_{k-1} = \frac{\rho_0}{\tau_{k-1}^2} \geq \frac{\rho_0(k+1)^2}{4}$ and $\rho_{k-1} S_{\rho_{k-1}}(z^k, \hat{\lambda}^0) \leq \rho_0 R_0^2$. Substituting these estimates into (6), we obtain (9). $\qquad\square$

### 4.1 Lower bound of convergence rate for the semi-strongly convex case

We consider again example (32), where we assume that $g$ is $\mu_g$-strongly convex. Algorithm 2 for solving (32) are special cases of (33) if $g$ is strongly convex. Then, by [28, Theorem 2], the lower bound complexity of (33) to achieve $\hat{x}$ such that $F(\hat{x}) - F^\star \leq \varepsilon$ is $\Omega\left(\frac{1}{\sqrt{\varepsilon}}\right)$. Consequently, the rate of Algorithm 2 stated in Theorem 3.2 is optimal.

## E  Additional numerical experiments

We provide more numerical examples to support our theory presented in the main text.

### 5.1 The $\ell_1$-Regularized Least Absolute Derivation (LAD)

We consider the following $\ell_1$-regularized least absolute derivation (LAD) problem widely studied in the literature:

$$F^\star := \min_{y \in \mathbb{R}^{p_2}} \left\{ F(y) := \|By - c\|_1 + \kappa \|y\|_1 \right\}, \tag{51}$$

where $B \in \mathbb{R}^{n \times \hat{p}}$ and $c \in \mathbb{R}^n$ are given, and $\kappa > 0$ is a regularization parameter. This problem is completely nonsmooth. If we introduce $x := By - c$, then we can reformulate (51) into (1) with two objective functions $f(x) := \|x\|_1$ and $g(y) := \kappa \|y\|_1$ and a linear constraint $-x + By = c$.

We use problem (51) to verify our theoretical results presented in Theorem 3.1 and Theorem 3.2. We implement Algorithm 1 (`NEAPAL`), its parallel scheme (`NEAPAL-par`), and Algorithm 2 (`scvx-NEAPAL`). We compare these algorithms with ASGARD [23] and its restarting variant, Chambolle-Pock's method [3], and standard ADMM [2]. For ADMM, we reformulate (51) into the following constrained setting:

$$\min_{x,y,z} \left\{ \|x\|_1 + \kappa \|z\|_1 \ | \ -x + By = c, \ y - z = 0 \right\}$$

to avoid expensive subproblems. We solve the subproblem in $x$ using a preconditioned conjugate gradient method (PCG) with at most 20 iterations or up to $10^{-5}$ accuracy.

We generate a matrix $B$ using standard Gaussian distribution $\mathcal{N}(0,1)$ without and with correlated columns, and normalize it to get unit column norms. The observed vector $c$ is generated as $c := Bx^\natural + \hat{\sigma}\mathcal{L}(0,1)$, where $x^\natural$ is a given $s$-sparse vector drawn from $\mathcal{N}(0,1)$, and $\hat{\sigma} = 0.01$ is the variance of noise generated from a Laplace distribution $\mathcal{L}(0,1)$. For problems of the size $(m,n,s) = (2000, 700, 100)$, we tune to get a regularization parameter $\kappa = 0.5$.

We test these algorithms on two problem instances. The configuration is as follows:

- For `NEAPAL` and `NEAPAL-par`, we set $\rho_0 := 5$, which is obtained by upper bounding $\frac{2\|\lambda^\star\|}{\|B\|\|y^0 - y^\star\|}$ as suggested by the theory. Here, $y^\star$ and $\lambda^\star$ are computed with the best accuracy using an interior-point algorithm in MOSEK.
- For `scvx-NEAPAL` we set $\rho_0 = \frac{1}{4\|B\|^2}$ by choosing $\mu_g = 0.5$.
- For Chambolle-Pock's method, we run two variants. In the first variant, we set step-sizes $\tau = \sigma = \frac{1}{\|B\|}$, and in the second one we choose $\tau = 0.01$ and $\sigma = \frac{1}{\|B\|^2 \tau}$ as suggested in [3], and it works better than $\tau = \frac{1}{\|B\|}$. We name these variants by `CP` and `CP-0.01`, respectively.
- For ADMM, we tune different penalty parameters and arrive at $\rho = 10$ that works best in this experiment.

The result of two problem instances are plotted in Figure 4. Here, `ADMM-1` and `ADMM-10` stand for ADMM with $\rho = 1$ and $\rho = 10$, respectively. `CP` and `CP-0.01` are the first and second variants of Chambolle-Pock's method, respectively. `ASGARD-rs` is a restarting variant of ASGARD, and `avg-` stands for the relative objective residuals evaluated at the averaging sequence in Chambolle-Pock's method and ADMM. Note that the $\mathcal{O}\left(\frac{1}{k}\right)$-rate of these two methods is proved for this averaging sequence.

Figure 4: Convergence behavior of 9 algorithmic variants on two instances of (51) after 1000 iterations. Left: Without correlated columns; Right: With $50\%$ correlated columns.

We can observe from Figure 4 that `scvx-NEAPAL` is the best. Both `NEAPAL` and `NEAPAL-par` have the same performance in this example and slightly slower than `CP-0.01`, `ADMM-10` and `ASGARD-rs`. Note that ADMM requires to solve a linear system by PCG which is always slower than other methods including `NEAPAL` and `NEAPAL-par`. `CP-0.01` works better than `CP` in late iterations but is slow in early iterations. `ASGARD` and `ASGARD-rs` remain comparable with `CP-0.01`. Since both Chambolle-Pock's method and ADMM have $\mathcal{O}\left(\frac{1}{k}\right)$-convergence rate on the averaging sequence, we also evaluate the relative objective residuals and plot them in Figure 4. Clearly, this sequence shows its $\mathcal{O}\left(\frac{1}{k}\right)$-rate but this rate is much slower than the last iterate sequence in all cases. It is also much slower than `NEAPAL` and `NEAPAL-par`, where both schemes have a theoretical guarantee.

## 5.2 Image compression using compressive sensing

In this last example, we consider the following constrained convex optimization model in compressive sensing of images:

$$\min_{Y \in \mathbb{R}^{p_1 \times p_2}} \left\{ f(Y) := \|\mathcal{D}Y\|_{2,1} \mid \mathcal{L}(Y) = b \right\}, \tag{52}$$

where $\mathcal{D}$ is 2D discrete gradient operator representing a total variation (isotropic) norm, $\mathcal{L} : \mathbb{R}^{p_1 \times p_2} \to \mathbb{R}^n$ is a linear operator obtained from a subsampled transformation scheme [2], and $b \in \mathbb{R}^n$ is a compressive measurement vector [1]. Our goal is to recover a good image $Y$ from a small amount of measurement $b$ obtained via a model-based measurement operator $\mathcal{L}$. To fit into our template (1), we introduce $x = \mathcal{D}Y$ to obtain two linear constraints $\mathcal{L}(Y) = b$ and $-x + \mathcal{D}Y = 0$. In this case, the constrained reformulation of (52) becomes

$$F^\star := \min_{x,Y} \left\{ F(z) := \|x\|_{2,1} \mid x - \mathcal{D}Y = 0, \ \mathcal{L}(Y) = b \right\},$$

where $f(x) = \|x\|_{2,1}$, and $g(Y) = 0$.

We now apply Algorithm 1 (`NEAPAL`), its parallel variant (`NEAPAL-par`), and Algorithm 2 (`scvx-NEAPAL`) to solve this problem and compare them with the `CP` method in [3] and ADMM [2]. We also compare our methods with a line-search variant `Ls-CP` of `CP` recently proposed in [3].

In `CP` and `Ls-CP`, we tune the step-size $\tau$ and find that $\tau = 0.01$ works well. The other parameters of `Ls-CP` are set as in the previous examples. For `NEAPAL` and `NEAPAL-par`, we use $\rho_0 := 2\|\mathcal{B}\|^2$. We also use $\rho_0 := 10\|\mathcal{B}\|^2$ and call the variant of Algorithm 1 and its parallel scheme `NEAPAL-v2` and `NEAPAL-par-v2`, respectively in this case. We set $\mu_g := \frac{1}{2\|\mathcal{B}\|}$ in `scvx-NEAPAL` as a guess for

restricted strong convexity parameter. For the standard `ADMM` algorithm, we tune its penalty parameter and find that $\rho := 20$ works best.

We test all the algorithms on 4 MRI images: `MRI-of-knee`, `MRI-brain-tumor`, `MRI-hands`, and `MRI-wrist`.[3] We follow the procedure in [2] to generate the samples using a sample rate of 25%. Then, the vector of measurements $c$ is computed from $c := \mathcal{L}(Y^\natural)$, where $Y^\natural$ is the original image.

Table 2: Performance and results of 8 algorithms on 4 MRI images

| Algorithms | $f(Y^k)$ | $\frac{\|\mathcal{L}(Y^k)-b\|}{\|b\|}$ | Error | PSNR | Time[s] | $f(Y^k)$ | $\frac{\|\mathcal{L}(Y^k)-b\|}{\|b\|}$ | Error | PSNR | Time[s] |
|---|---|---|---|---|---|---|---|---|---|---|
| | MRI-knee $(779 \times 693)$ | | | | | MRI-brain-tumor $(630 \times 611)$ | | | | |
| NEAPAL | 24.350 | 2.637e-02 | 4.672e-02 | 83.93 | 80.15 | 36.101 | 2.724e-02 | 6.575e-02 | 79.50 | 53.77 |
| NEAPAL-par | 24.335 | 2.539e-02 | 4.676e-02 | 83.93 | 98.38 | 36.028 | 2.738e-02 | 6.595e-02 | 79.47 | 52.71 |
| NEAPAL-v2 | 28.862 | 7.125e-05 | 4.143e-02 | 84.98 | 73.56 | 39.317 | 5.226e-05 | 6.310e-02 | 79.85 | 52.97 |
| NEAPAL-par-v2 | 29.183 | 7.247e-05 | 4.007e-02 | 85.27 | 95.49 | 39.594 | 5.338e-05 | 6.258e-02 | 79.93 | 51.64 |
| scvx-NEAPAL | 24.633 | 2.295e-02 | 4.424e-02 | 84.41 | 87.96 | 36.783 | 2.184e-02 | 5.780e-02 | 80.62 | 65.12 |
| CP | 24.897 | 2.674e-02 | 4.629e-02 | 84.01 | 101.22 | 37.745 | 3.613e-02 | 7.896e-02 | 77.91 | 63.71 |
| Ls-CP | 24.955 | 2.638e-02 | 4.659e-02 | 83.96 | 106.11 | 38.139 | 3.414e-02 | 7.485e-02 | 78.37 | 66.12 |
| ADMM | 25.071 | 2.556e-02 | 4.654e-02 | 83.97 | 902.79 | 38.941 | 2.895e-02 | 6.135e-02 | 80.10 | 655.81 |
| | MRI-hands $(1024 \times 1024)$ | | | | | MRI-wrist $(1024 \times 1024)$ | | | | |
| NEAPAL | 45.207 | 2.081e-02 | 2.765e-02 | 91.37 | 146.41 | 29.459 | 1.802e-02 | 3.224e-02 | 90.04 | 152.51 |
| NEAPAL-par | 45.207 | 2.081e-02 | 2.765e-02 | 91.37 | 140.41 | 29.459 | 1.802e-02 | 3.224e-02 | 90.04 | 148.12 |
| NEAPAL-v2 | 48.679 | 7.336e-05 | 2.074e-02 | 93.87 | 138.65 | 30.578 | 8.516e-05 | 2.572e-02 | 92.00 | 146.05 |
| NEAPAL-parallel-v2 | 48.858 | 7.483e-05 | 2.008e-02 | 94.15 | 148.79 | 30.768 | 8.766e-05 | 2.473e-02 | 92.34 | 146.64 |
| scvx-NEAPAL | 45.426 | 1.820e-02 | 2.588e-02 | 91.95 | 154.35 | 29.403 | 1.647e-02 | 3.131e-02 | 90.29 | 157.35 |
| CP | 45.723 | 2.489e-02 | 3.895e-02 | 88.40 | 159.74 | 30.052 | 2.032e-02 | 3.661e-02 | 88.93 | 165.58 |
| Ls-CP | 53.640 | 2.724e-02 | 3.924e-02 | 88.33 | 162.94 | 39.396 | 2.353e-02 | 3.856e-02 | 88.48 | 168.29 |
| ADMM | 45.985 | 2.034e-02 | 3.443e-02 | 89.47 | 1691.53 | 29.922 | 1.825e-02 | 3.686e-02 | 88.88 | 1503.56 |

The performance and results of these algorithms are summarized in Table 2, where $f(Y^k) := \|\mathcal{D}Y^k\|_{2,1}$ is the objective value, `Error` $:= \frac{\|Y^k - Y^\natural\|_F}{\|Y^\natural\|_F}$ presents the relative error between the original image $Y^\natural$ to the reconstruction $Y^k$ after $k = 300$ iterations.

We observe the following facts from the results of Table 2.

- `NEAPAL`, `NEAPAL-par`, and `scvx-NEAPAL` are comparable with `CP` in terms of computational time, PSNR, objective values, and solution errors.
- `NEAPAL-v2` and `NEAPAL-par-v2` give better PSNR and solution errors, but have slightly worse objective value than the others.
- `Ls-CP` is slower than our methods due to additional computation.
- `ADMM` gives similar result in terms of the objective values, solution errors, and PSNR, but it is much slower than other methods due to the PCG inner loop.

## Footnotes

[3]These images are from https://radiopaedia.org/cases/4090/studies/6567 and https://www.nibib.nih.gov