[Reviews · NeurIPS 2018]

Reviewer 1



Summary: The paper proposed 4 new variants of Augmented Lagrangian methods, which called NAPALA (non-ergodic alternating proximal augmented Lagrangian algorithms) to solve non-smooth constrained convex optimization problems under different problem structure assumptions. The first algorithm only requires f and g to be convex but neither necessarily strongly convex nor smooth. Especially, its parallel variant (8) has more advantages. The second algorithm requires either f or g to be strongly convex. Its variant (12) also allows to have parallel steps. The author(s) could derive the convergence rate at the last iterate (non-ergodic), which are the state-of-the-art results. Numerical experiments also show good overall performance of the proposed algorithms compared to other algorithms. Comments: In general, I really like the results of the paper since it could achieve the convergence rate at the last iterate (non-ergodic sense), which is very useful for sparse and low rank optimization or image processing. In the non-strongly convex case, there exist many algorithms that can achieve O(1/k) convergence rate, but such a rate is often in an ergodic sense. Note that ergodic sequences possibly break the sparsity and low rankness in sparse and low-rank optimization, respectively. Especially, in image processing, ergodic sequences may destroy the sharp edge structure of images. Although the convergence rate of ADMM (O(1/k)) and its accelerated versions is widely studied in the literatures, this paper proposes 4 novel algorithms which I have not seen in the literature. Compared to existing methods in the literature, the parallel variant (8) seems to have several advantages while achieving O(1/k)-non-ergodic optimal rate under only convexity and strong duality. Algorithm 2 allows using different sequence to achieve either ergodic or nonergodic rate on y, but nonergodic rate on x. It also linearizes the y-subproblem compared to ADMM. I believe that the results of the paper are non-trivial and have significant theoretical contribution to optimization and machine learning community. Besides the above mentioned positive points, I also have the following comments: 1) I saw that you are able to achieve the convergence rate of O(1/k^2) in the case that either f or g is strongly convex (section 4.1). However, when you do parallel step, you need f and g are strongly convex (together) to have the rate O(1/k^2). What do you think about the convergence rate when either f or g is strongly convex in this parallel step? Do you think it is still possible to achieve O(1/k^2)? Have you tried to do experiments to see the behavior? 2) Could you please explain more clearly about Section 4.3? What should we do if f and/or g are smooth? 3) A minor comment: I think your algorithms could also expand to coordinate descent. What is your opinion about this?

Reviewer 2



This paper shows an optimal rate for the non-ergodic sequence governed by the Augmented Lagrangian. This is theoretically an interesting result. The numerical illustration for the square-root LASSO looks to be good but in the square-root elastic net problem, it seems that the Chambolle-Pock decreases stably while NAPALA seems to be flatter.

Reviewer 3



This paper is basically a traditional optimization paper, which introduces a new method (similar, but not equivalent, to admm) for a class of problems (convex with linear constraints). Convergence is provided for the base case, plus a parallel and two strong convexity. variants. The convergence rates are not too surprising (1/k for general convex, 1/k2) if either f or g are strongly convex) The experiments are moderately large data. (not trivial, not surprisingly large). A weakness of this paper is that it is not all that well motivated, in that we can think of many methods that can accomplish similar tasks with similar per iteration costs (method of multipliers, douglas rachford, pock chambolle). I believe this method has some advantages over each of these, and it would benefit the paper if these were discussed in the introduction. In particular, it seems the observed performance is much better than chambolle pock, so it would be good to have some discussion as to why that might be (is it problem specific? commonly observed? Theoretically verified?) Also the proofs are pretty involved and rather hard to verify. It might be nice for clarification purposes to have a table of variables, or some other way of easily checking the steps.